# DNA damage signaling in *Drosophila* macrophages modulates systemic cytokine levels in response to oxidative stress

**Fabian Hersperger[1,2], Tim Meyring[1], Pia Weber[1], Chintan Chhatbar[1], Gianni Monaco[1,3], Marc S Dionne[4,5], Katrin Paeschke[6,7], Marco Prinz[1,8,9], Olaf Groß[1,8,9], Anne-Kathrin Classen[10,11], Katrin Kierdorf[1,8,11]***

[1]Institute of Neuropathology, Faculty of Medicine, Medical Center, University of Freiburg, Freiburg, Germany; [2]Faculty of Biology, University of Freiburg, Freiburg, Germany; [3]Institute for Transfusion Medicine and Gene Therapy, Medical Center-University of Freiburg, Freiburg, Germany; [4]MRC Centre for Molecular Bacteriology and Infection, Imperial College London, London, United Kingdom; [5]Department of Life Sciences, Imperial College London, London, United Kingdom; [6]Department of Oncology, Haematology and Rheumatology, University Hospital Bonn, Bonn, Germany; [7]Institute of Clinical Chemistry and Clinical Pharmacology, University Hospital Bonn, Bonn, Germany; [8]Center for Basics in NeuroModulation (NeuroModulBasics), Faculty of Medicine, University of Freiburg, Freiburg, Germany; [9]Signalling Research Centres BIOSS and CIBSS, University of Freiburg, Freiburg, Germany; [10]Hilde-Mangold-Haus, Faculty of Biology, University of Freiburg, Freiburg, Germany; [11]CIBSS-Centre for Integrative Biological Signalling Studies, University of Freiburg, Freiburg, Germany

***For correspondence:**
katrin.kierdorf@uniklinik-freiburg.de

**Competing interest:** The authors declare that no competing interests exist.

**Abstract** Environmental factors, infection, or injury can cause oxidative stress in diverse tissues and loss of tissue homeostasis. Effective stress response cascades, conserved from invertebrates to mammals, ensure reestablishment of homeostasis and tissue repair. Hemocytes, the *Drosophila* blood-like cells, rapidly respond to oxidative stress by immune activation. However, the precise signals how they sense oxidative stress and integrate these signals to modulate and balance the response to oxidative stress in the adult fly are ill-defined. Furthermore, hemocyte diversification was not explored yet on oxidative stress. Here, we employed high-throughput single nuclei RNA-sequencing to explore hemocytes and other cell types, such as fat body, during oxidative stress in the adult fly. We identified distinct cellular responder states in plasmatocytes, the *Drosophila* macrophages, associated with immune response and metabolic activation upon oxidative stress. We further define oxidative stress-induced DNA damage signaling as a key sensor and a rate-limiting step in immune-activated plasmatocytes controlling JNK-mediated release of the pro-inflammatory cytokine *unpaired-3*. We subsequently tested the role of this specific immune activated cell stage during oxidative stress and found that inhibition of DNA damage signaling in plasmatocytes, as well as JNK or upd3 overactivation, result in a higher susceptibility to oxidative stress. Our findings uncover that a balanced composition and response of hemocyte subclusters is essential for the survival of adult *Drosophila* on oxidative stress by regulating systemic cytokine levels and cross-talk to other organs, such as the fat body, to control energy mobilization.

## eLife assessment

This study elucidates the role of a specific hemocyte subpopulation in oxidative damage response by establishing connections between DNA damage response and the JNK-JAK/STAT axis to regulate energy metabolism. The identification of this distinct hemocyte subpopulation through single-cell RNA sequencing analysis and the finding of hemocytes that respond to oxidative stress are **important**. The method for single-cell RNA sequencing and related analyses are **convincing** and experiments linking oxidative stress to DNA damage and energy expenditure are **solid**. The finding of stress-responsive immune cells capable of influencing whole-body metabolism adds insights for cell biologists and developmental biologists in the fields of immunology and metabolism.

## Introduction

Metazoan organisms must constantly adapt to the surrounding environment. Environmental factors, including radiation, pathogens or toxins, permanently lead to the generation of reactive oxygen species (ROS) (*Pham-Huy et al., 2008*). In vertebrates and invertebrates, low levels of ROS support cellular pathways like immune cell differentiation and even immune function including antimicrobial defense (*Sinenko et al., 2011*; *Herb and Schramm, 2021*). At high concentrations, ROS lead to oxidative stress, a deleterious process that negatively impacts protein synthesis and stability, lipid metabolism as well as overall genome stability. ROS-mediated cellular damage can further result in degenerative diseases or cancer (*Nathan and Cunningham-Bussel, 2013*). Therefore, metazoan organisms employ efficient repair and defense mechanisms to circumvent high levels of oxidative stress and subsequent organ damage. Hence, keeping the balance between the production and the antioxidant-dependent clearance of ROS is essential for the survival and well-being of the affected organism (*Finkel and Holbrook, 2000*). Oxidative stress induced tissue damage is most often followed by a strong local immune response. These molecular processes are conserved across vertebrates and invertebrates such as the fruit fly *Drosophila melanogaster* (*Moghadam et al., 2021*).

Hemocytes, the *Drosophila* blood cells, are essential cellular components of the immune and stress response machinery. They consist of three different subtypes: crystal cells, lamellocytes, and plasmatocytes (*Meister and Lagueux, 2003*). Plasmatocytes are considered to be macrophage equivalents in *Drosophila* and comprise up to 95% of all hemocytes in the adult fly (*Banerjee et al., 2019*). They share many functions as well as the multilayered ontogeny with vertebrate tissue macrophages (*Gold and Brückner, 2014*; *Holz et al., 2003*). Adult plasmatocytes are actively involved in many immune functions such as pathogen recognition, defense, and phagocytosis but also chronic immune activation upon metabolic stress by dietary lipids (*Péan et al., 2017*; *Woodcock et al., 2015*). Recent studies highlighted the essential role of adult plasmatocytes as key players in regulation of tissue homeostasis, including the gut by regulating the intestinal stem cell niche or in muscles by controlling glucose homeostasis (*Kierdorf et al., 2020*; *Ayyaz et al., 2015*).

Similar to higher eukaryotes, levels of ROS are used as a central antimicrobial defense mechanism of hemocytes against invading pathogens (*Myers et al., 2018*). Moreover, larval hemocytes respond differentially to ROS, with two cell subsets mounting a biphasic immune reaction to invading bacteria by controlling hemocyte spreading and adhesion as well as crystal cell rupture (*Myers et al., 2018*). Larval hemocytes have been also attributed to respond to increased oxidative stress of other tissues, as seen in response to oxidative stress in the eye-antenna imaginal disc (*Fogarty et al., 2016*). Similarly, by regulating intestinal stem cell (ISC) proliferation in the damaged midgut during infection- or chemical-induced oxidative stress (*Ayyaz et al., 2015*), adult hemocytes were described to release the bone morphogenetic protein (BMP) homologue *decapentaplegic* (*dpp*) to facilitate tissue repair (*Ayyaz et al., 2015*). Other studies pointed to an essential role of hemocyte-derived *unpaired-3* (*upd3*), which is the *Drosophila* orthologue of Interleukin-6, in the regulation of ISC proliferation during septic injury (*Chakrabarti et al., 2016*), and wound healing during sterile injury (*Chakrabarti and Visweswariah, 2020*). In line with these findings, hemocytes have been implicated in organ-to-organ immune signaling for example upon infection-induced injury in the gut and antimicrobial peptide (AMP) production in the fat body (*Wu et al., 2012*). Hemocyte activation is thus a hallmark of oxidative stress in defined tissues, although their role in modulating and controlling oxidative stress response to facilitate tissue repair and survival of the organism remains undefined.

Here, we demonstrate an essential role of hemocytes to control susceptibility to oxidative stress, where hemocytes orchestrate systemic oxidative stress response by defined cellular states identified by unbiased transcriptomic profiling. Plasmatocytes segregate in oxidative stress responder subclusters, characterized by immune activation and cellular stress response, and potential bystander subsets associated with metabolic gene regulations. Mechanistically, we unravel that the direct immune response to oxidative stress by plasmatocytes is driven by c-Jun N-terminal kinase (JNK) and Jak/STAT pathway activation with subsequent induction of the pro-inflammatory cytokine *upd3*. This inflammatory cascade is to a certain amount regulated by DNA damage signaling in plasmatocytes and essentially defines the susceptibility of the fly to oxidative stress. Deficiencies in the DNA damage signaling machinery in plasmatocytes lead to elevated JNK activation, cytokine release and subsequently a higher susceptibility of the adult fly to oxidative stress. Hence, our data point to a key role of plasmatocytes in balancing systemic cytokine levels and stress response upon oxidative stress. Collectively, we provide new insights on the role of macrophages in orchestrating systemic stress responses in other tissues and to balance tissue wasting and energy mobilization.

## Results

### Hemocytes control susceptibility to oxidative stress

In vertebrates and invertebrates, prolonged oxidative stress can cause severe tissue damage, hence controlled activation of stress signaling cascades are essential for the survival of the organism and the resolution of oxidative stress with subsequent tissue repair. Yet the precise role of macrophages in the initiation and control of oxidative stress signaling needs to be determined.

To analyze the role of macrophages in signaling cascades to oxidative stress, we took advantage of the well-established Paraquat (PQ)-feeding model in adult *Drosophila melanogaster* to induce systemic oxidative stress (*Wang et al., 2003*). Feeding of 7 days old *w;Hml-Gal4,UAS-2xeGFP/+* (*Hml/+*) flies with PQ-supplemented sucrose solution led to an increased lethality of adult *Drosophila* after 18 hr in a dose-dependent manner (0 mM: 1.25 ± 1.25%, 2 mM: 2.5 ± 1.6 4%, 15 mM: 27.78 ± 5.47%, and 30 mM: 38.75 ± 6.67%) (*Figure 1—figure supplement 1A*). In agreement with previous studies (*Wang et al., 2003*), 15 mM PQ was chosen for further experiments (*Figure 1A*). First, we determined which systemic changes are associated with susceptibility to oxidative stress. Infection-induced damage in enterocytes can result in gut leakage and subsequent death (*Chakrabarti et al., 2016*). We tested if the death of flies after 18 hr on 15 mM PQ food was associated with PQ-induced damage of gut enterocytes and resulting gut leakage, but we found no indication of gut leakage on 15 mM PQ food (*Figure 1—figure supplement 1B*). Only very few flies on control food and PQ indicated gut leakage of the food color to thorax and abdomen, which was unrelated to the treatment and not altered between the groups (control: 3.75%, 2 mM PQ: 2.53 %, 15 mM PQ: 1.11% and 30 mM PQ: 3.75 %) (*Figure 1—figure supplement 1B*). We investigated if aberrant systemic induction of immune signaling cascades are seen upon PQ-mediated oxidative stress. For this, gene expression levels of different known immune activated genes were analyzed by qPCR such as *unpaired* (*upd*) cytokines (*upd1*, *upd2*, *upd3*), *Jak/STAT* down-stream targets (*Turandot-A* (*TotA*), *suppressor of cytokine signaling at 36E* (*Socs36E*)), stress induced JNK activation (*puckered* (*puc*)) (*Figure 1B*), as well as expression of *nuclear factor 'kappa-light-chain-enhancer' of activated B-cells* (*Nfkb*)-induced antimicrobial peptides (AMP) and further cytokines such as *eiger*, *dpp*, and *dawdle* (*daw*) (*Figure 1—figure supplement 1C–D*). Our analyses did not reveal significant differences for these gene groups on a systemic level. In contrast, genes associated with carbohydrate metabolism such as *insulin receptor* (*InR*), *phosphoenolpyruvate carboxykinase* (*Pepck*), and *Thor* were significantly upregulated upon PQ-feeding (*Figure 1C*). These findings point to an activation of *forkhead box, sub-group O* (*foxo*)-target genes and a potentially reduced insulin signaling, despite the lack of significant alterations in the expression of the Insulin-like peptides (*Ilps*) *Ilp2*, *Ilp3,* and *Ilp5* (*Figure 1C*).

To test the consequences for energy mobilization and nutrient storage, we measured free glucose content, as well as glycogen and triglyceride levels in whole flies after 18 hr on control or 15 mM PQ. Although no changes in free glucose levels (control: 1.49±0.02 µg and PQ: 1.50±0.07 µg, ns p=0.954) (*Figure 1D*) were found, we observed a clear decrease in stored glycogen (control: 1.02±0.07 µg and PQ: 0.33±0.08 µg, **p=0.003) (*Figure 1E*) and stored triglycerides (*Figure 1F*) in PQ-treated flies compared to control. Next, we examined histological sections of flies to define which tissues might be

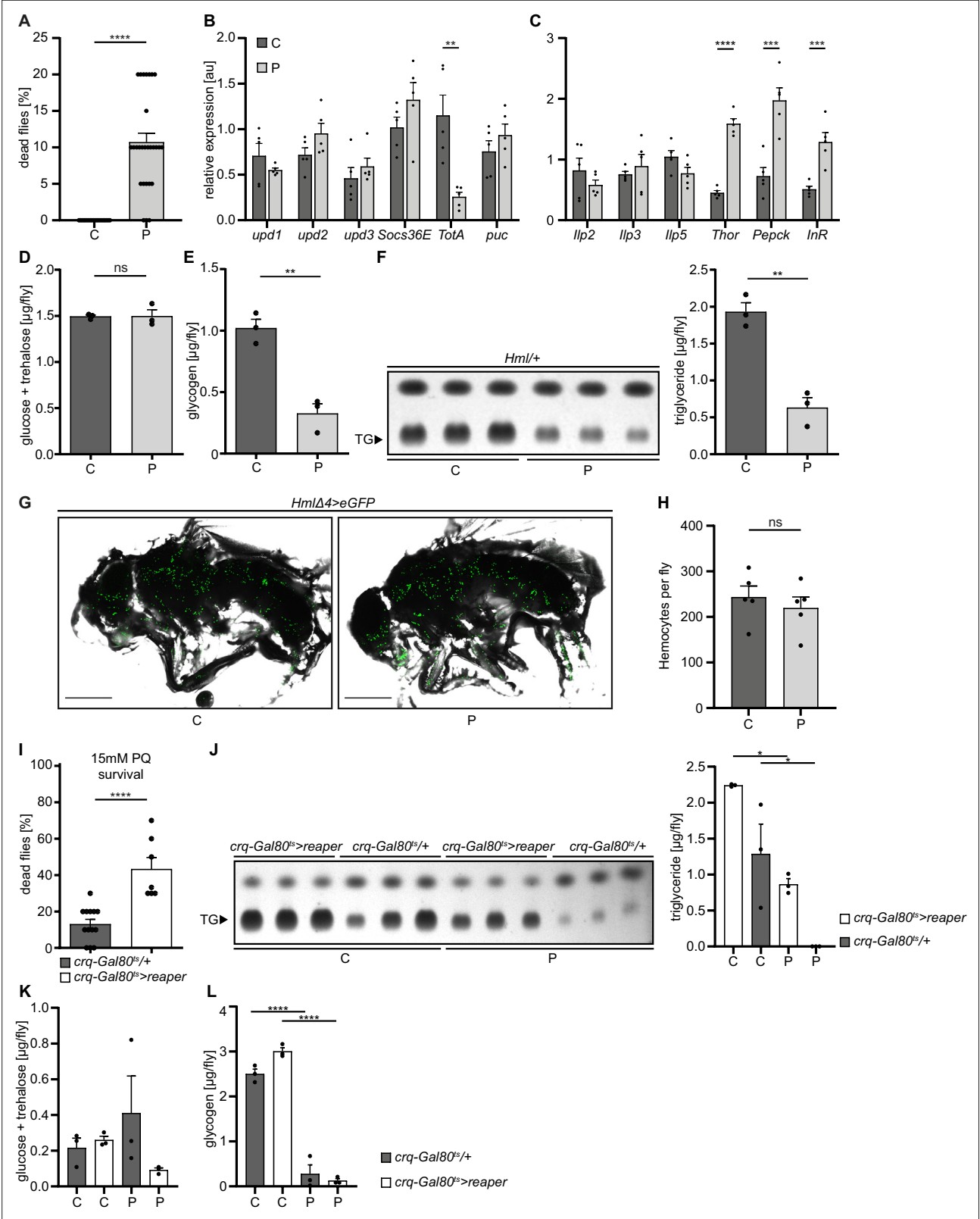

**Figure 1.** Loss of hemocytes increases the susceptibility of adult *Drosophila* to oxidative stress by Paraquat feeding. (**A**) Survival of *Hml/+* flies with control food (C) (n=20) and 15 mM Paraquat (P) (n=28) at 29 °C. Five independent experiments were performed. Student's unpaired t-test: ****p<0.0001, each data point represents a sample with 10 flies. (**B–C**) RT-qPCR of whole flies were performed to investigate gene-expressions of the JAK-STAT-signaling pathway (**B**) and insulin-signaling pathway (**C**). All transcript levels were normalized to the expression of the loading control *Rpl1*

*Figure 1 continued on next page*

*Figure 1 continued*

and are shown in arbitrary units [au]. Each data point (n=5) represents a sample of three whole flies. Student's unpaired t-tests were performed for each transcript to compare the gene-expression between control and treated flies: **p<0.01; ***p<0.001; ****p<0.0001. (**D**) Glucose and (**E**) glycogen levels of whole flies were measured in control and PQ-treated flies. Each dot (n=3) represents a sample containing three flies. Student's unpaired t-test: **p<0.01. (**F**) Triglyceride (TG) amounts were determined via thin-layer chromatography (TLC). Left panel: Representative image of a TLC plate, each band represents a sample with ten flies. Right panel: Quantification thereof. n=3 per group were analyzed. Student's unpaired t-test: **p<0.01. (**G**) Representative images of 7 days old *HmlΔ4>eGFP* flies treated with 5% sucrose solution and 15 mM PQ, respectively. Scale bars = 500 µm. (**H**) Hemocyte quantification of 7 days old *HmlΔ4>eGFP* flies treated with control food (5% sucrose solution) (n=5) or 15 mM Paraquat (n=5). Each data point represents one fly. Statistical significance was tested using student's unpaired t-test: p=0.508. (**I**) Survival of *crq-Gal80ts/+* control flies (n=13) and hemocyte depleted *crq-Gal80ts>reaper* flies (n=7) on control food (n=20) and 15 mM Paraquat (n=28) at 29 °C. Student's unpaired t-test: ****p<0.0001, each data point represents a sample with 10 flies. (**J**) TG amounts in *crq-Gal80ts/+* control flies and hemocyte-deficient *crq-Gal80ts>reaper* flies on control food or Paraquat food. Left panel: Representative image of a TLC plate, each band represents a sample with ten flies. Right panel: Quantification thereof. n=3 per group were analyzed. One-way ANOVA: *p<0.05. (**K**) Glucose and (**L**) Glycogen levels of whole flies were measured in *crq-Gal80ts/+* control flies and hemocyte-deficient *crq-Gal80ts>reaper* flies. Each dot (n=3) represents a sample with three whole flies. One-way ANOVA: ****p<0.0001.

The online version of this article includes the following figure supplement(s) for figure 1:

**Figure supplement 1.** Paraquat-induced oxidative stress does not cause gut leakage, but detrimental changes in triglyceride storage in the fat body.

affected. Hematoxylin and eosin (H&E) and Oil Red O (ORO) stainings showed a shrinkage in fat body cell size and stored lipid-filled vesicles in the abdominal fat body compared to flies kept under control conditions *Figure 1—figure supplement 1E*. Fat body development, energy mobilization and storage have been recently connected to the presence, cell number and subsequent cross-talk of hemocytes with fat body (*Cox et al., 2021*), therefore we quantified GFP+ hemocytes in *Hml/+* flies after 18 hr on PQ or control food (*Figure 1G–H*). Our analysis did not reveal alterations in hemocyte numbers and especially no oxidative stress induced loss of hemocytes (control: 243.4±24.33 hemocytes per fly and PQ: 219.6±24.22 hemocytes per fly, ns p=0.508) (*Figure 1G–H*). To evaluate if hemocytes are essential contributors to the stress-mediated metabolic changes and susceptibility to oxidative stress, we used 'hemocyte-deficient' *crq-Gal80ts>reaper* (*w;tub-Gal80ts/UAS-CD8-mCherry;crq-Gal4/ UAS-reaper*) flies and tested their survival rate compared to control flies on PQ food (*Figure 1I*). Hemocyte-deficient flies displayed a significantly higher mortality rate on 15 mM PQ (43.33 ± 6.30%) compared to control flies (13.08 ± 2.63%) (*Figure 1I*). Next, we analyzed triglycerides, as well as free glucose and glycogen levels in whole flies after 18 hr on control or 15 mM PQ. Interestingly, we found that hemocyte-deficient flies compared to the control genotype mobilized less triglycerides from the fat body during oxidative stress (*Figure 1J*). When we looked into free glucose levels, we found that these flies show a decrease in free glucose upon PQ feeding, whereas the control genotype showed similar free glucose levels as on normal food (*Figure 1K*). In contrast, we did not detect any difference in glycogen levels between the two genotypes on control food and on PQ food (*Figure 1L*). Our findings show detrimental changes in energy mobilization upon oxidative stress in the absence of profound systemic overactivation of immune signaling cascades, however the presence or absence of hemocytes seem to be a key determinant of the susceptibility to oxidative stress and results in altered energy mobilization in adult *Drosophila*. This strongly points to an essential role for hemocytes in controlling stress response and organismal survival upon oxidative stress.

## Paraquat exposure generates diverse transcriptional states of adult hemocytes with a specific cell state associated with oxidative stress response

To gain further insights into the alterations induced in the adult fly upon oxidative stress and especially to the hemocyte compartment, we decided to take advantage of unbiased single nucleus RNA-sequencing (snRNA-seq). We isolated nuclei from control food and PQ-treated *Hml-DsRed.nuc* flies with a modified Frankenstein protocol (*G Martelotto, 2020*) and performed fluorescence activated nuclear sorting of DAPI+Draq7+ nuclei (*Figure 2—figure supplement 1A*). From two independent sorts, we obtained a total of 29,000 nuclei, which were utilized for subsequent droplet-based barcoding of single nuclei, and generation of single nucleus cDNA libraries. We could successfully obtain gene expression data from 11,839 nuclei (*Figure 2—figure supplement 1B–D*). Quality control analysis followed by unsupervised hierarchical clustering of all analyzed nuclei using Seurat package

(*Hao et al., 2021*) revealed 20 distinct nuclei clusters (C) with differential gene expression (*Figure 2A*). Cell identity was annotated according to the cell type-specific signatures determined by the Fly Cell Atlas consortium (*Figure 2A*; *Li et al., 2022*). All annotated cell types were detected in PQ-treated and control flies (*Figure 2B*).

Using cell type-specific signatures, three of the clusters were annotated as hemocytes (clusters: 2, 11, and 14; *Figure 2A–B*). Re-scaling and clustering of all 1354 identified hemocyte nuclei revealed 8 distinct clusters (*Figure 2C*). We verified hemocyte-specific signature genes across these clusters including genes such as the transcription factor *Serpent* (*Srp*) and one or more of the following hemocyte signature genes: *Hemolectin* (*Hml*), *croquemort* (*crq*), *Nimrod C1* (*NimC1*), *eater,* and *Peroxidasin* (*Pxn*) (*Figure 2—figure supplement 2A–L*; *Cattenoz et al., 2020*; *Tattikota et al., 2020*). Cluster 8 expressed various crystal cell-specific markers including *lozenge* (*lz*), *pebbled* (*peb*), *prophenoloxidase 1* (*PPO1*), and *prophenoloxidase 2* (*PPO2*), whereas all other clusters seemed to be plasmatocytes (*Figure 2—figure supplement 2A–L*). In line with previous reports, the embryonic and larval hemocyte markers *Hemese* (*He*) (data not shown) and *singed (sn)* were not expressed across any of the identified clusters (*Figure 2—figure supplement 2H*). Interestingly, when we defined which hemocyte clusters were enriched in control flies or PQ-treated flies, the two plasmatocyte clusters C1 and C7 were primarily derived from control flies, whereas C2, C3, C4, and C6 are predominantly found in PQ-treated flies (*Figure 2D* and *Figure 2—figure supplement 2M*). The plasmatocyte cluster C5 as well as the crystal cell cluster C8 presented in both conditions to similar proportions (*Figure 2D* and *Figure 2—figure supplement 2M*).

Next, we analyzed the TOP20 differentially expressed (DE) genes across all hemocyte clusters. We found clear differences between the seven distinct plasmatocyte clusters and the crystal cell cluster (*Figure 2E* and *Supplementary file 1*). The crystal cell cluster C8 showed a high expression of crystal cell specific genes including *lz*, *peb*, *PPO1*, and *PPO2* (*Figure 2E*). Plasmatocyte cluster C1, which was mostly present in control conditions, had a remarkable expression of hemocyte specific marker genes such as *Hml* and *Pxn,* as well as migration-associated genes including the semaphorins *Sema2a and Sema5c* (*Figure 2E*). Yet this cluster also showed higher expression of genes contributing to metabolic regulations such as the L-gulonate 3-dehydrogenase *Had2* and the gluconolactonase *regucalcin* (*Figure 2E*). The cluster C7 and C1, mostly found in control flies, substantially shared gene expression profiles. In contrast, cluster C2 and C3 were nearly exclusively found in PQ-treated flies and showed expression of genes involved in metabolic adaptations. Due to this finding, we verified that these cell clusters are *bona fide* hemocytes and not fat body cells. We analyzed the expression of characteristic fat body signature genes, such as *Trehalose-6-phosphate synthase 1* (*Tps1*), *CG16758*, *CG2233*, *CG4716*, *Ultrabithorax* (*Ubx*), *Calcium/calmodulin-dependent protein kinase I* (*CaMKI*), *Desaturase 1* (*Desat1*), *Phosphoribosyl pyrophosphate synthetase* (*Prps*), *CG34166* and *Apolipoprotein lipid transfer particle* (*Apoltp*) across all identified hemocyte clusters (*Figure 2—figure supplement 3A*). Metabolic adaptation within these clusters included high expression of the *glucose transporter 4 enhancer factor* (*Glut4EF*) and the *phosphatidate phosphatase Lipin* (*Lpin*) in C2 or the induced expression of the lipase *brummer* (*bmm*), the *autophagy-associated gene-17* (*Atg17*) or the *lipophorin receptor 1* (*LpR1*), involved in uptake of neutral lipids from the hemolymph, in C3 (*Figure 2E*). Overall gene expression profiles in clusters C2 and C3 showed an induction of insulin signaling, increased autophagy and lipolysis pointing to adaptation of energy mobilization in these clusters upon PQ treatment (*Figure 2E*). Cluster C4 was characterized by a remarkable upregulation of genes involved in translation including many ribosomal proteins such as *RpL8*, *RpL31*, or *RpS13* (*Figure 2E*). Furthermore, C4 included genes associated with unsaturated fatty acid metabolism such as *Desaturase 1 (Desat1)*. Similarly, cluster C5 also showed induction of genes involved in fatty acid metabolism such as *ELOVL fatty acid elongase 7* (*ELOVL*) (*Figure 2E*). Of note, we also found *Tie-like receptor tyrosine kinase* (*Tie*) specifically expressed in this cluster, which is described to bind PDGF- and VEGF-related factor (Pvf) ligands and contribute to cell survival and migration, as well as actin-associated binding proteins including *polychaetoid* (*pyd*) and *formin 3* (*form3*). All these gene expression changes are pointing to a plasmatocyte cluster with a gene expression profile associated with mobility and migration. Among all plasmatocyte clusters, we found cluster C6 which presented a strong immune activation phenotype with induction of Jak/STAT target genes including *Fasciclin 3* (*Fas3*) and *Turandot-M* (*TotM*) among its TOP20 DE genes, but also activation of oxidative stress response genes such as the heat shock

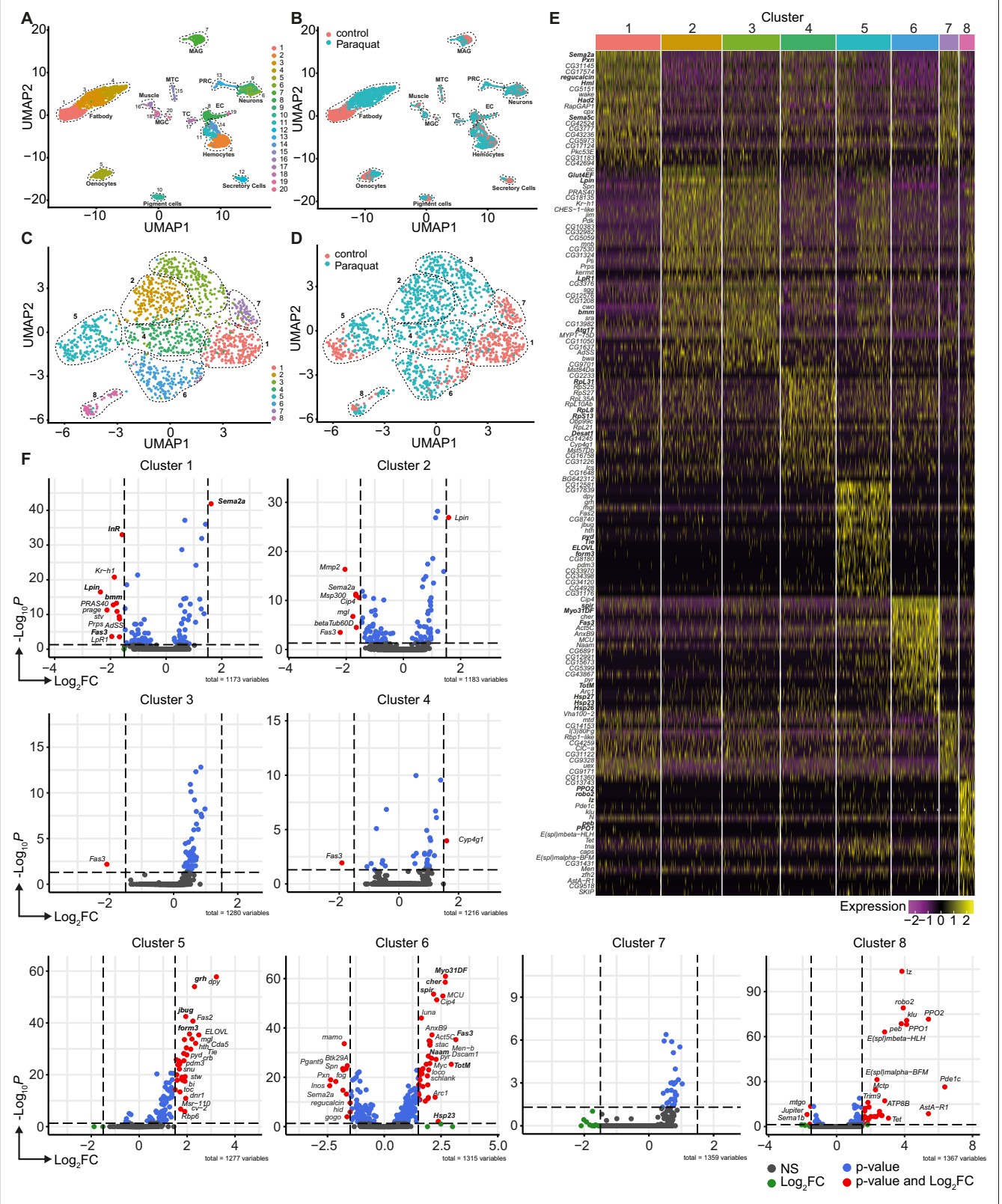

**Figure 2.** Unbiased single nuclei transcriptomic profiling identified diverse transcriptomic states of hemocytes associated with oxidative stress response. (**A**) Uniform Manifold Approximation and Projection (UMAP) visualization of single nucleus states from flies exposed to 15 mM PQ containing or control food. Dashed lines indicate the broad cell types. Male accessory gland main cells (MAG), Malpighian tubule principal cells (MTC), male germline cells (MGC), outer photoreceptor cells (PRC), tracheolar cells (TC), epithelial cells (EC). Colors indicate distinct clusters. n=11,839 nuclei are shown. (**B**) UMAP

*Figure 2 continued*

visualization of nuclei from (**A**) colored by treatment. Nuclei from control group are labeled in red and nuclei from PQ group are labeled in blue. Dashed lines indicate the broad cell type as (**A**). (**C**) UMAP visualization of hemocytes from (**A**) after sub-setting and re-clustering. Colors and dashed lines indicate distinct clusters. n=1354 nuclei are shown. (**D**) UMAP visualization of hemocyte clusters colored by treatment. Nuclei from control group are labeled in red and nuclei from PQ group are labeled in blue. Dashed lines indicate distinct clusters. n=1354 nuclei are shown. (**E**) Heat map of top 20 DE genes for each hemocyte cluster. Gene names are indicated on the left. Colors in the heat map correspond to normalized scaled expression. (**F**) Volcano plots comparing pseudo bulk gene expression of individual hemocyte cluster vs. all hemocytes. The –$\log_{10}$-transformed adjusted P value (P adjusted, y-axis) is plotted against the $\log_2$-transformed fold change (FC) in expression between the indicated cluster vs remaining hemocytes (x-axis). Non-regulated genes are shown in grey, not significantly regulated genes are shown in green, significantly regulated genes with a FC <1.5 are shown in blue and significantly regulated genes with a FC >1.5 are shown in red.

The online version of this article includes the following figure supplement(s) for figure 2:

**Figure supplement 1.** Gating strategy for the FACS isolation of nuclei from control and PQ-treated flies.

**Figure supplement 2.** Specific signature genes identify plasmatocyte and crystal cell clusters within the eight identified hemocyte clusters.

**Figure supplement 3.** Fat body signature genes are highly expressed across the eight distinct fat body cell clusters and are absent in hemocyte cell clusters.

proteins *Hsp23*, *Hsp26* and *Hsp27* or cytoskeletal genes including *myosin Myo31DF* or *spire* (*spir*) (*Figure 2E*).

Due to the partially shared expression profile of genes across hemocyte clusters, we wondered if there were genes exclusively expressed in individual clusters. However, cluster C1 showed that *Sema2a* seems to be exceptionally high expressed in these homeostatic plasmatocytes and genes associated with metabolic regulation (e.g. *InR*, *Lpin* or *bmm*) or Jak/STAT activation (e.g. *Fas3*), as seen in PQ-associated clusters, are downregulated (*Figure 2F*). Cluster C5 showed enrichment for genes such as the transcription factor *grainy head* (*grh*) but also cytoskeletal-associated genes such as *jitterbug* (*jbug*) and *form3,* suggesting a migratory potential of plasmatocytes from this cluster within their host tissues (*Figure 2F*). Interestingly, plasmatocyte cluster C6 presented with the most profound gene expression changes compared to all other clusters. Similar to our previous observations, we identified that genes specifically expressed in this cluster were associated with Jak/STAT pathway activation (e.g. *Fas3*, *TotM*), cytoskeletal rearrangements and migration (e.g. *Myo31DF*, *spir* or *cheerio* [*cher*]) and response to oxidative stress (e.g. *Nicotinamide amidase* [*Naam*] or *Hsp23*; *Figure 2F*). We also included the crystal cell cluster C8 in this analysis and verified unique expression of crystal cell markers compared to the seven plasmatocyte clusters (*Figure 2F*). Our snRNA-seq analysis revealed that adult hemocytes respond differentially to oxidative stress with a specific cluster of plasmatocytes associated with immune activation and stress response. Despite our findings that on systemic level we did not detect a strong induction of immune activation upon oxidative stress, we now provide evidence that plasmatocyte cluster C6 showed a substantial immune response toward oxidative stress.

## Direct response to oxidative stress *in vitro* mimics the transcriptional changes seen in cluster C6 plasmatocytes *in vivo*

Plasmatocytes responded to PQ-mediated oxidative stress in different cellular states. We defined cluster C6 as a cell state, which is associated with oxidative stress response and immune activation. Since cluster C6 showed gene expression associated with oxidative stress and immune activation, we focused our further analysis on this cluster. To further substantiate our findings and address the signaling pathways that could lead to diverse transcriptional states, we inferred the transcription factor (TF) network based on gene expression data (*Figure 3A*). This analysis revealed that genes with binding sites for *Xbp1* were enriched. *Xbp1* is a transcription factor associated with stress response. In addition to this, the Nf-κB-related transcription factor *Relish* (*Rel*) and *kayak* (*kay*), a component of the *activator protein-1* (*AP-1*) transcription factor complex induced by JNK signaling, were also enriched in this cluster (*Figure 3A*). To determine if the gene expression signature of cluster C6 plasmatocytes reflects a functional state of plasmatocytes directly induced by oxidative stress, we treated S2 cells, a *Drosophila* plasmatocyte cell line, with different concentrations of PQ (15 mM and 30 mM) *in vitro* (*Figure 3—figure supplement 1*). First, we performed flow cytometric analysis of ROS levels via CellROX staining after 6 and 24 hr PQ treatment compared to controls (*Figure 3—figure supplement 1A–B*). We observed a dose-dependent increase in intracellular ROS at all time points analyzed with higher ROS levels at 6 hr (0 mM: 1903±64 median fluorescence intensity (MFI), 15 mM:

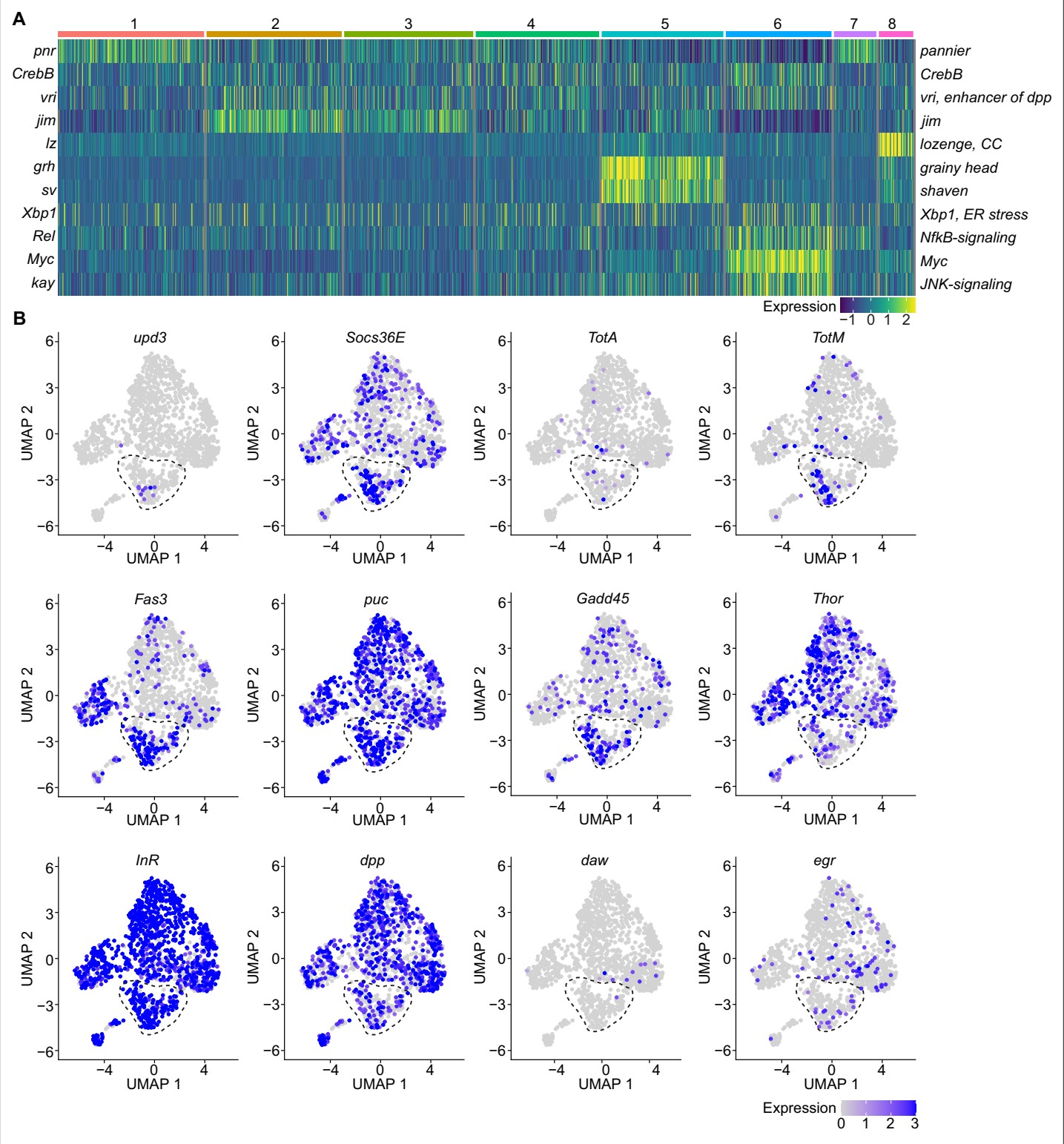

**Figure 3.** A specific cluster of plasmatocytes responds to oxidative stress with immune activation by Jak/STAT, DDR, and JNK signaling. (**A**) Key transcription factors (TFs) regulating hemocyte states. Heat map of scaled TF activity at a single-cell level in hemocyte states from *Figure 2C*. Colors in the heat map correspond to scaled expression. Numbers on top indicate the cluster identity. Labels on the left and right indicate the TFs. (**B**) Feature plots showing scaled expression of selected genes associated with JAK/STAT signaling (*upd3, Socs36E, TotA, TotM, Fas3*), JNK and DNA damage signaling (*puc, Gadd45*), insulin signaling (*Thor, InR*), TGFβ signaling (*dpp, daw*), and TNF signaling (*egr*).

The online version of this article includes the following figure supplement(s) for figure 3:

*Figure 3 continued on next page*

*Figure 3 continued*

**Figure supplement 1.** PQ treatment of S2 cells *in vitro* induces reactive oxygen species, immune activation and DNA damage.

6735±1,096 MFI, and 30 mM: 7435±237 MFI) compared to 24 hr (0 mM: 1236±135 MFI, 15 mM: 4456±228 MFI, and 30 mM: 5557±303 MFI; *Figure 3—figure supplement 1C*). Increase in ROS and oxidative stress can lead to cellular damage such as damage in nuclear and mitochondrial DNA.

We next tested if plasmatocytes directly exposed to oxidative stress might also experience this type of stress response by performing comet assays of treated and untreated S2 cells. Upon PQ treatment, S2 cells showed a remarkable and dose-dependent increase in DNA damage compared to untreated cells (0 mM: 101.0±2.1, 5 mM: 117.6±3.3, 10 mM: 136.6±3.8, and 15 mM: 147.8±5.5, data shown as olive tail moment; *Figure 3—figure supplement 1D*). We were then wondering if PQ-treated S2 cells would resemble cluster C6 plasmatocytes *in vivo* and performed gene expression analysis of S2 cells treated with 15 mM PQ after 6 and 24 hr. Comparable to our results for cluster C6 *in vivo*, we found a profound induction of the Jak/STAT target genes by qRT-PCR. In S2 cells, we also detected a significant increase of the Jak/STAT cytokines *upd1*, *upd2,* and *upd3* (*Figure 3—figure supplement 1E*), which we did not see before *in vivo*. Furthermore, we detected an increased expression of the JNK-target gene *puckered* (*puc*), but barely any induction of metabolic genes (*Figure 3—figure supplement 1E*). All these results further support our hypothesis that direct exposure of S2 cells with PQ *in vitro* mimics gene expression changes seen in a particular activated cell stage of plasmatocytes upon oxidative stress *in vivo*.

Because we found enrichment of genes associated with the Nf-κB-related transcription factor *Rel* in the immune-activated plasmatocyte cluster C6, we further verified if any other cytokines or AMPs are induced in PQ-treated S2 cells. We identified a slight, but significant, induction of the cytokine *daw*, but no other alterations in *dpp* or the AMPs *Drs*, *Dro*, or *Mtk* (*Figure 3—figure supplement 1F*). The profound induction of Jak/STAT target genes together with the activation of JNK target genes and oxidative stress induced DNA damage in S2 cells *in vitro* highly support our hypothesis that cluster C6 plasmatocytes reflect a cellular state associated with the direct response to oxidative stress *in vivo*. Along this line, we went back to our snRNA-seq data and specifically searched for Jak/STAT target genes (*upd3*, *Socs36E*, *TotA*, *TotM,* and *Fas3*) across all clusters (*Figure 3B*). Here, we identified an enriched expression of these genes in cluster C6, and a unique expression of the pro-inflammatory cytokine *upd3* in cluster C6, which was not detected in any other hemocyte cluster (*Figure 3B*). Of note, we did not detect elevated *upd3* levels in gene expression profiling of whole flies (*Figure 1B*), but now unraveled elevated *upd3* induction in the particular hemocyte cell cluster C6 upon PQ treatment. Interestingly, we also identified that *Growth arrest and DNA damage-inducible 45* (*Gadd45*), a gene linking DNA damage signaling and stress induced JNK signaling, was enriched in plasmatocyte cluster C6 (*Figure 3B*). This could indicate that plasmatocytes within cluster C6 undergo oxidative stress mediated DNA damage, which results in activation of the DNA damage repair machinery but also JNK activation, similar to what we have seen in S2 cells *in vitro*. However within our snRNA-seq analysis, the JNK target gene *puc* was expressed across all hemocyte clusters analyzed *in vivo* without a specific induction in cluster C6 (*Figure 3B*). We looked again into the expression of metabolic genes including *Thor* and *InR* as well as other cytokines such as *dpp*, *daw*, or *egr* but could not detect a specific enrichment in cluster C6 similar to our results from PQ-treated S2 cells *in vitro* (*Figure 3B*). These findings from the snRNA-seq analysis *in vivo* and S2 cells *in vitro* support the hypothesis that cluster C6 plasmatocytes represent a functional state of activated plasmatocytes directly responding to oxidative stress by JNK signaling and Jak/STAT pathway activation. Furthermore, we found evidence that cluster C6 specifically expresses the cytokine *upd3*, which could reflect an important mediator for oxidative stress response.

## Oxidative stress induces distinct cell states in the fat body, including the induction of a Jak/STAT responsive and the loss of an AMP-producing cell cluster

The diversification of plasmatocytes during oxidative stress and the defined cytokine producing immune-activated cell cluster C6 could play a key role in immune response to oxidative stress and also for interorgan communication. To further explore this hypothesis, we aimed to gain further insights

into single-cell transcriptomic profiles of the *Drosophila* fat body during oxidative stress. Using cell type-specific signatures, three of the clusters were annotated as fat body cells (clusters: 1, 3, and 4; *Figure 2A–B*). Re-scaling and clustering of all 3150 identified fat body nuclei revealed 8 distinct clusters (*Figure 4A*). First, we verified fat body-specific signature gene expression across these clusters (*Figure 2—figure supplement 3B*) and excluded expression of hemocyte-specific signature genes (*Figure 2—figure supplement 3C*). From the eight distinct fat body cell clusters, we identified that cluster C1, C6 and C8 were mainly enriched with cells obtained from control flies, whereas clusters C2-5 were almost exclusively derived from PQ-treated flies (*Figure 4B*, *Figure 2—figure supplement 3D*). Only cluster C7 represents cells from both conditions (*Figure 4B*, *Figure 2—figure supplement 3D*). To define the transcriptional changes between these eight fat body cell clusters, we analyzed the TOP20 DE genes across all fat body cell clusters (*Figure 4C*, *Supplementary file 2*), as well as the unique regulated genes in each specific cluster compared to all others (*Figure 4D*). Here the cluster C1, mostly derived of fat body cells from control flies, showed an enrichment for gene induction associated with triglyceride storage, such as *Fatty acid synthase 1* (*FASN1*), *Glycerol-3-phosphate dehydrogenase 1* (*Gpdh1*) or *Acyl-CoA synthetase long-chain* (*Acsl*), as well as fat body signature genes (*Figure 4C–D*). Interestingly cluster C8, which mostly consisted of cells from control food, showed a distinct expression of AMP genes including *Dro*, *Drs* or *Diptericin-B* (*DptB*), however this cluster completely disappeared after PQ feeding (*Figure 4C–D*). In contrast, PQ treatment induced several fat body-specific cell clusters which were not found in control conditions. For example, cluster C2, C4, and C5 showed a strong gene induction of oxidative stress response genes, but also genes associated with Jak/STAT pathway (e.g. *Socs36E*, *upd3*, *Fas3*, *TotA*) and JNK activation (e.g. *puc*, *kay*) (*Figure 4C–D*). In all of these clusters, genes associated with triglyceride storage were significantly reduced. This is in line with our findings showing a reduction in triglyceride storage in the fat body upon oxidative stress (*Figure 1F*, *Figure 1—figure supplement 1E*). Overall, our data revealed that the fat body has a unique heterogeneity upon oxidative stress with a clear separation of clusters with JNK and Jak/STAT activation signature, which could be either activated by the PQ-induced oxidative stress itself or by the cytokines released by immune-activated hemocytes, especially *upd3*.

## Loss of DNA damage signaling in adult hemocytes results in elevated *upd3* levels and a decreased survival on oxidative stress

To evaluate the role of immune activation in hemocytes during oxidative stress response, we aimed to specifically explore the implication of DNA damage signaling in hemocytes (*Figures 1–3*). Previously, it has been observed that oxidative stress-induced DNA damage supports immune activation in macrophages via DNA damage signaling (*Morales et al., 2017*). Based on our findings, we predict that also in adult hemocytes oxidative stress not only drives genome instability and activates repair, but plays a role in the induction and modulation of immune activation via specific gene expression changes in cluster C6 plasmatocytes. To test if DNA damage signaling is involved in the response of adult hemocytes to oxidative stress, we analyzed flies with a hemocyte-specific knock-down for different genes of the DNA damage repair (DDR) machinery including the DNA damage sensing kinases *telomere fusion* (*tefu*) and *meiotic 41* (*mei-41*) and the MRN complex protein *nbs*. Of note, by targeting adult hemocytes, mostly plasmatocytes are targeted which make up 95% of all hemocytes in the adult fly.

We generated *Hml >mei-41-IR* (*w;Hml-Gal4,UAS-2xeGFP/UAS-mei-41-IR*), *Hml >tefu* IR (*w;Hml-Gal4,UAS-2xeGFP/UAS-tefu-IR*), *Hml >mei41 IR;tefu-IR* (*w;Hml-Gal4,UAS-2xeGFP/UAS-mei-41-IR;UAS-tefu-IR*) and *Hml >nbs* IR (*w;Hml-Gal4,UAS-2xeGFP/UAS-nbs-IR*) flies (*Figure 5—figure supplement 1A*). First, we compared their susceptibility to PQ treatment after 18 hr and found significantly increased susceptibility to the treatment across all DDR-deficient lines compared to the control genotype (*Hml/+*: $10.7 \pm 1.2\%$, *Hml >mei-41-IR*: $40.7 \pm 4.8\%$, *Hml >tefu-IR*: $20.7 \pm 4.3\%$, *Hml >mei41 IR;tefu-IR*: $37.5 \pm 4.8\%$ and *Hml >nbs-IR*: $48.2 \pm 6.7\%$) (*Figure 5A*). In these flies, decreased survival after PQ did not correlate to an overall reduction in lifespan on normal food and a decrease in fitness (*Figure 5—figure supplement 1B*).

Next, we inquired if DDR-deficiency in hemocytes might increase susceptibility of the fly to other stresses. For this we challenged all genotypes with starvation. Importantly, we did not observe a decreased survival of DDR-deficient flies on starvation compared to the control genotype, indicating that the observed susceptibility to oxidative stress of flies with a DDR-deficiency in hemocytes is

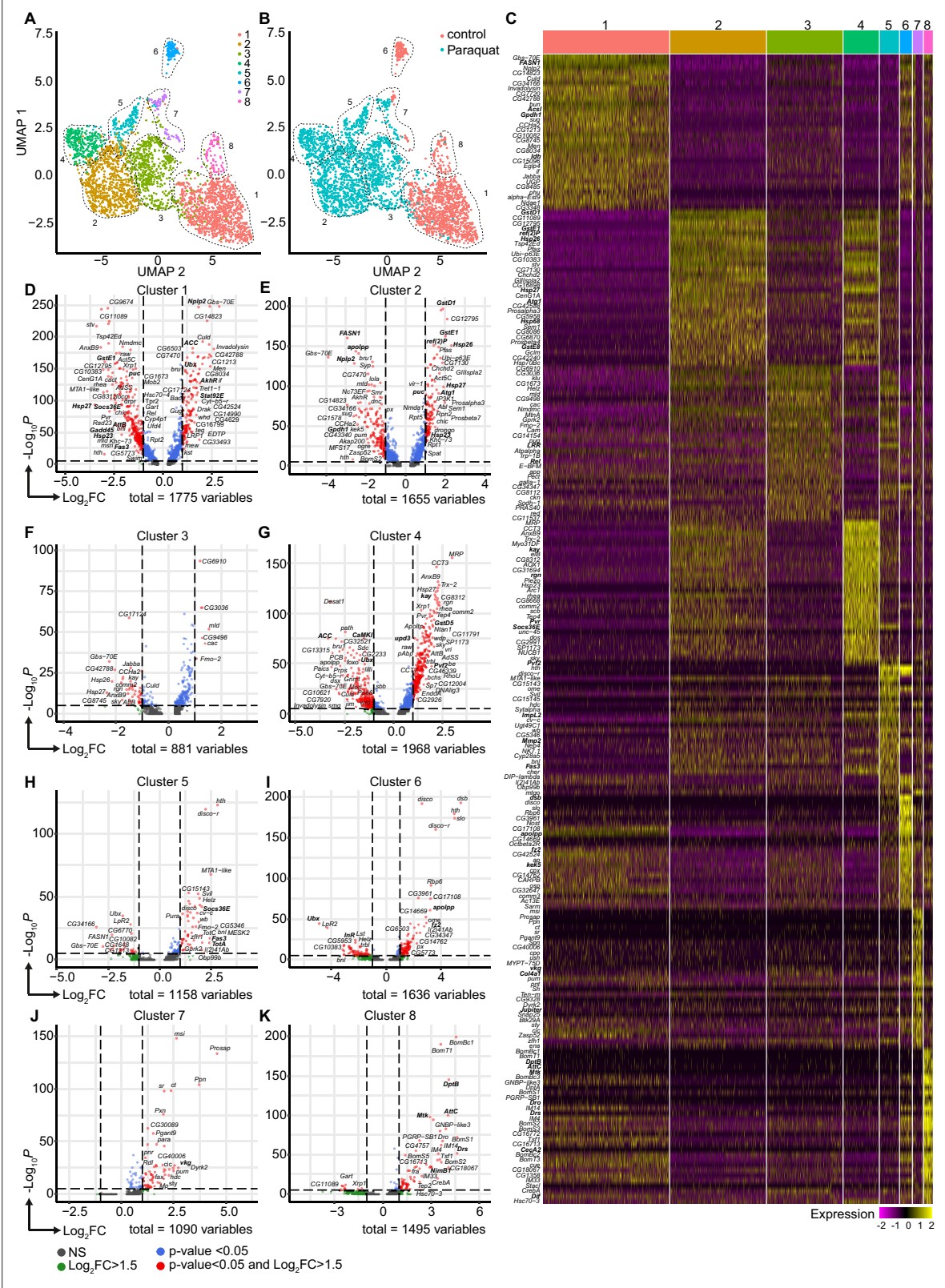

**Figure 4.** Oxidative stress induced different transcriptomic states in fat body cells, including cells with a distinct Jak/STAT activation signature. (**A**) UMAP visualization of fat body cells from **Figure 2A** by unsupervised clustering. Colors and dashed lines indicate different clusters. Each dot represents one nucleus. n=3150 nuclei are shown. (**B**) UMAP of fat body cell clusters split by treatment. Nuclei from control flies are labeled in red and nuclei from PQ treated flies are labeled in blue. Dashed lines indicate different clusters. Each dot represents one nucleus. n=3150 nuclei are shown. (**C**) Heat map of

*Figure 4 continued on next page*

*Figure 4 continued*

top 20 DE genes for each fat body cluster. Gene names indicated on the left. Fold change of gene expression is color coded as indicated in legend. (**D-K**) Volcano plots comparing pseudo bulk gene expression of individual fat body cluster vs. all fat body cells. The –log$_{10}$-transformed adjusted p-value (p adjusted, y-axis) is plotted against the log$_2$-transformed fold change (FC) in expression between the indicated cluster vs remaining hemocytes (x-axis). Non-regulated genes are shown in grey, not significantly regulated genes are shown in green, significantly regulated genes with a FC <1.5 are shown in blue and significantly regulated genes with a FC >1.5 are shown in red.

specifically caused by oxidative stress due to PQ treatment (**Figure 5—figure supplement 1C**). As defects in the DDR machinery in hemocytes might already affect the development during embryonic and larval stages, we used a hemocyte-specific temperature-inducible knock-down of *mei-41*, *tefu* and *nbs1* which was induced by a temperature shift to 29 °C directly after eclosion from the pupae to induce a knockdown in adult hemocytes only. We analyzed the lifespan of *Hml-Gal80$^{ts}$>mei-41-IR* (*w;Hml-Gal4,UAS-2xeGFP/UAS-mei-41-IR;tub-Gal80$^{ts}$*), *Hml-Gal80$^{ts}$>tefu* IR (*w;Hml-Gal4,UAS-2xeGFP/UAS-tefu-IR;tub-Gal80$^{ts}$*), *Hml-Gal80$^{ts}$>mei-41-IR;tefu-IR* (*w;Hml-Gal4,UAS-2xeGFP/UAS-mei-41-IR;tub-Gal80$^{ts}$/UAS-tefu-IR*) and *Hml-Gal80$^{ts}$>nbs* IR (*w;Hml-Gal4,UAS-2xeGFP/UAS-nbs-IR;tub-Gal80$^{ts}$*) compared to control flies *Hml-Gal80$^{ts}$/+* (*w;Hml-Gal4,UAS-2xeGFP/+;tub-Gal80$^{ts}$/+*) and again found no alterations in overall lifespan of all genotypes analyzed on control food except a slight decrease in lifespan of *Hml-Gal80$^{ts}$>mei-41-IR;tefu-IR*. (**Figure 5—figure supplement 1D**). In contrast, we again found a higher susceptibility to oxidative stress via PQ-feeding in the DDR-deficient lines compared to control (*Hml-Gal80$^{ts}$/+*: 30.3 ± 5.2%, *Hml-Gal80$^{ts}$>mei-41-IR*: 36.2 ± 4.3%, *Hml-Gal80$^{ts}$>tefu-IR*: 52.4 ± 6.9%, *Hml-Gal80$^{ts}$>mei-41-IR;tefu-IR*: 72.3 ± 6.3% and *Hml-Gal80$^{ts}$>nbs-IR*: 36.2 ± 4.3%) (**Figure 5—figure supplement 1E**), which directly confirms our previous results and conclusions (**Figure 5A**).

As we have seen before that loss of hemocytes is detrimental for the survival on oxidative stress, we wanted to exclude a reduction in hemocytes in the DDR-deficient fly lines as a cause of the higher susceptibility to oxidative stress. We quantified GFP$^+$ hemocytes in *Hml>mei-41-IR*, *Hml>tefu*-IR, *Hml>mei-41-IR;tefu-IR* and *Hml>nbs*-IR flies (**Figure 5B**, **Figure 5—figure supplement 1A**). We did not find alteration in hemocyte number across all fly lines on control food (with the exception of an increase in hemocytes in the *Hml>tefu*-IR line), nor did we observe any effect of PQ treatment on the number of hemocytes in DDR-deficient flies (**Figure 5B**, **Figure 5—figure supplement 1A**).

Next, we analyzed if the loss of the DDR signaling machinery via knockdown of *tefu* and *mei41* results in increased DNA damage in hemocytes. Comet assays of isolated hemocytes from DDR-deficient flies and the control genotype on PQ and control food only indicated a slight increase in DNA damage in these flies, which demonstrates that the higher susceptibility of DDR-deficient flies is not due to a loss of hemocyte numbers or a substantial increase in DNA damage, but rather due to changes in the downstream DDR-signaling events modulating immune activation of hemocytes (**Figure 5C**). Other studies demonstrated that upon tissue damage JNK-mediated secretion of the JAK/STAT cytokines upd1, upd2 and upd3 is a critical step for innate immune signaling (**Jiang et al., 2009**; **Pastor-Pareja et al., 2008**). Furthermore, DNA damage signaling is essential to induce such an immune response and to regulate insulin signaling (**Karpac et al., 2011**). To elucidate the implication of DDR signaling on hemocyte activation and oxidative stress resistance, we measured the gene expression levels of *upd* cytokines as well as JNK and Jak/STAT targets in DDR-deficient flies and controls on PQ and control food. Loss of DNA damage signaling in hemocytes induced a significant induction of *upd3* expression upon PQ treatment in *Hml>tefu*-IR and *Hml>mei-41-IR;tefu-IR* flies (**Figure 5D**). In contrary, *upd1* and *upd2* were not significantly changed in DDR-deficient flies (**Figure 5D**). We measured the Jak/STAT target genes *Socs36E* and *TotA* on systemic level. A slight induction of *Socs36E* expression in all DDR-deficient genotypes on PQ, but only *Hml>mei-41-IR;tefu-IR* flies showed a significant induction of *Socs36E* upon PQ treatment compared to control food and *Hml/+* flies on PQ (**Figure 5D**). *TotA* expression tended to be reduced upon PQ treatment across all genotypes analyzed. However, when we checked the levels of *puc*, as a *JNK signaling* target gene, we found a significant induction on PQ treatment in *Hml >tefu* IR, *Hml >mei-41-IR;tefu-IR* and *Hml >nbs* IR flies, which was not seen in control flies upon PQ treatment (**Figure 5D**). When we analyzed the gene expression of genes involved in metabolism and energy mobilization, we detected a significant induction and dysregulation of *Thor* and *InR* in all DDR-deficient genotypes upon PQ treatment compared to control flies (**Figure 5E**), as well as a slight induction of *Pepck1* (**Figure 5E**). To exclude that these changes in energy mobilization

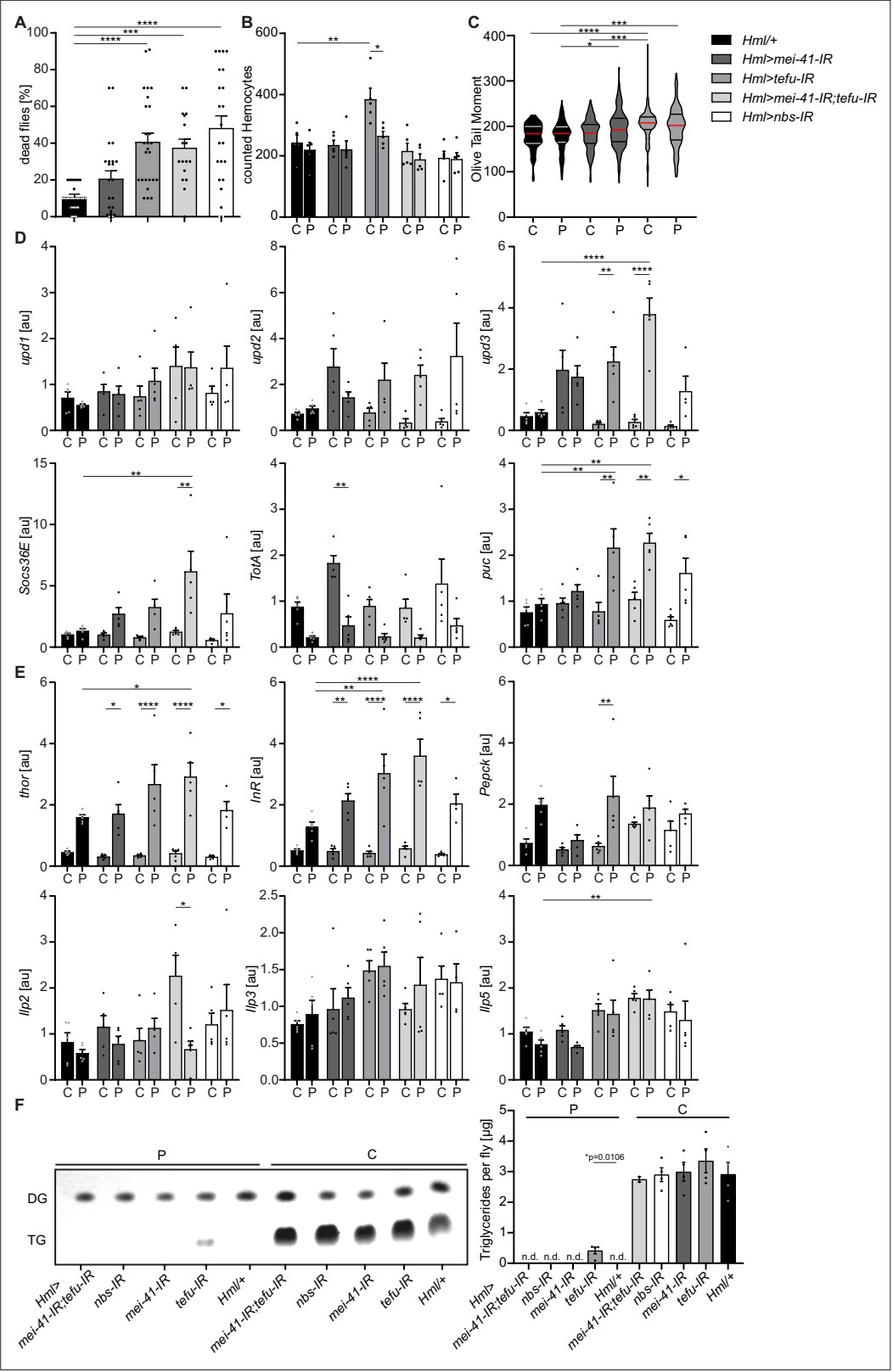

**Figure 5.** Loss of DNA damage signaling activity in hemocytes leads to an increase in systemic *upd3* levels and a higher susceptibility to oxidative stress. (**A**) Survival of *Hml/+* (n=28), *Hml >mei41 IR* (n=23); *Hml >tefu IR* (n=24); *Hml >mei41 IR,tefu-IR* (n=16) and *Hml >nbs* IR (n=22) flies on control food and PQ. Each data point represents a sample of 10 flies. Mean ± SEM is shown. Three independent experiments were performed. One-way ANOVA:

*Figure 5 continued on next page*

*Figure 5 continued*

***p<0.001; ****p<0.0001. (**B**) Hemocyte quantification of *Hml/+* control flies and DDR-knockdowns on control food (C) and 15 mM Paraquat (P). Each data point represents one fly (n=5 for all groups). Mean ± SEM is shown. One-way ANOVA: *Hml/+* (C) vs *Hml >tefu* IR (C), **p=0.0052; *Hml >tefu* IR (C) vs *Hml >tefu* IR (P), *p=0.03. (**C**) Comet assay of isolated hemocytes from *Hml/+* (C: n=73; P: n=129), *Hml >mei41* IR (C: n=119; P: n=132) and *Hml >tefu* IR (C: n=175; P: n=125) - flies on control food and PQ. Olive tail moment of comet assays of sorted hemocyte nuclei is shown. One-way ANOVA: *p<0.05; ***p<0.001; ****p<0.0001. (**D**) Gene expression analysis for *Jak/STAT* target genes via RT-qPCR in *Hml/+* control flies and DDR-knockdowns on control food (C) and 15 mM Paraquat (P). All transcript levels were normalized to the expression of the loading control *Rpl1* and are shown in arbitrary units [au]. Each data point (n=5) represents a sample of three individual flies. Mean ± SEM is shown. Two-way ANOVA: *p<0.05; **p<0.01; *** p<0.001; ****p<0.0001. (**E**) Gene expression analysis for insulin signaling target genes and *Ilps* via RT-qPCR in *Hml/+* control flies and DDR-knockdowns on control food (C) and 15 mM Paraquat (P). All transcript levels were normalized to the expression of the loading control *Rpl1* and are shown in arbitrary units [au]. Each data point (n=5) represents a sample of three individual flies. Mean ± SEM is shown. Two-way ANOVA: *p<0.05; **p<0.01; *** p<0.001; ****p<0.0001. (**F**) Triglyceride (TG) levels of *Hml/+*, *Hml >mei41* IR, *Hml >tefu* IR, *Hml >mei41* IR,tefu-IR and *Hml >nbs* IR flies on control food and PQ determined via thin-layer chromatography (TLC). One representative TLC is shown (left panel). Quantification thereof is shown on the right. Each data point (n=4 per group) represents a sample of 10 individual flies. Mean ± SEM is shown. n.d.=not detectable. One-way ANOVA was performed for both groups (15 mM PQ and control) separately: *p=0.0106.

The online version of this article includes the following figure supplement(s) for figure 5:

**Figure supplement 1.** Loss of DNA damage, upd3 or JNK signaling in hemocytes alters susceptibility to oxidative stress but not overall lifespan.

are due to altered expression of *Ilps*, we analyzed the levels of *Ilp2*, *Ilp3* and *Ilp5* in all genotypes on control and PQ treatment, but could not detect remarkable alterations here (***Figure 5E***). To further exclude that the higher susceptibility of DDR-deficient flies and seen alterations in stress response are due to inefficient mobilization of triglycerides, we measured stored triglycerides across all genotypes and found efficient mobilization of triglyceride storage upon PQ treatment in all genotypes compared to control food (***Figure 5F***).

From our results, we can conclude that DNA damage signaling in adult hemocytes, mostly consisting of plasmatocytes, essentially determines susceptibility to oxidative stress. Loss of DNA damage signaling in hemocytes results in upregulation of JNK signaling and elevated expression of the pro-inflammatory cytokine *upd3*, as well as transcriptional alterations in energy mobilization and insulin signaling. Based on our previous results on the role of hemocytes during oxidative stress, we can therefore assume that the DNA damage signaling machinery in hemocytes is activated by oxidative stress and plays an essential modulatory and rate-limiting role for subsequent immune activation.

## Adult hemocytes control susceptibility to oxidative stress by JNK signaling and *upd3* release

To this point, our data suggests that oxidative stress-induced DNA damage signaling in hemocytes limits immune activation and stress response upon oxidative stress. As we found that increased *upd3* levels correlate with a higher susceptibility to oxidative stress of flies with DDR-deficient hemocytes, we aimed to test if hemocyte-derived *upd3* is a major source to regulate susceptibility to PQ-induced oxidative stress. To do so, we either over-expressed *upd3* in hemocytes in *Hml>upd3* (*w;Hml-Gal4,UAS-2xeGFP/UAS-upd3*) flies or performed a hemocyte-specific knockdown of *upd3* in *Hml>upd3*-IR (*w;Hml-Gal4,UAS-2xeGFP/UAS-upd3-IR*) flies (***Figure 5—figure supplement 1F***). Both genotypes showed a normal overall lifespan compared to the *Hml/+* control (***Figure 5—figure supplement 1G***), but when we compared their susceptibility to PQ-induced oxidative stress, *Hml>upd3* flies showed a drastically decreased survival on PQ, whereas *Hml>upd3* IR flies did not show a changed survival on PQ treatment after 18 hr (***Figure 6A***). We tested *upd3* expression levels on a whole fly level and found a slight but not significant induction in *Hml/+* flies after PQ treatment compared to control food, in line with our previous results (***Figures 1B and 6B***). As expected, *Hml>upd3* flies showed increased levels of *upd3* in both treatment groups compared to the other genotypes, but also showed a significant increase of *upd3* upon PQ compared to control food (***Figure 6B***). Surprisingly, *Hml>upd3*-IR flies still showed a slight induction of *upd3* upon PQ treatment, which could potentially be derived from

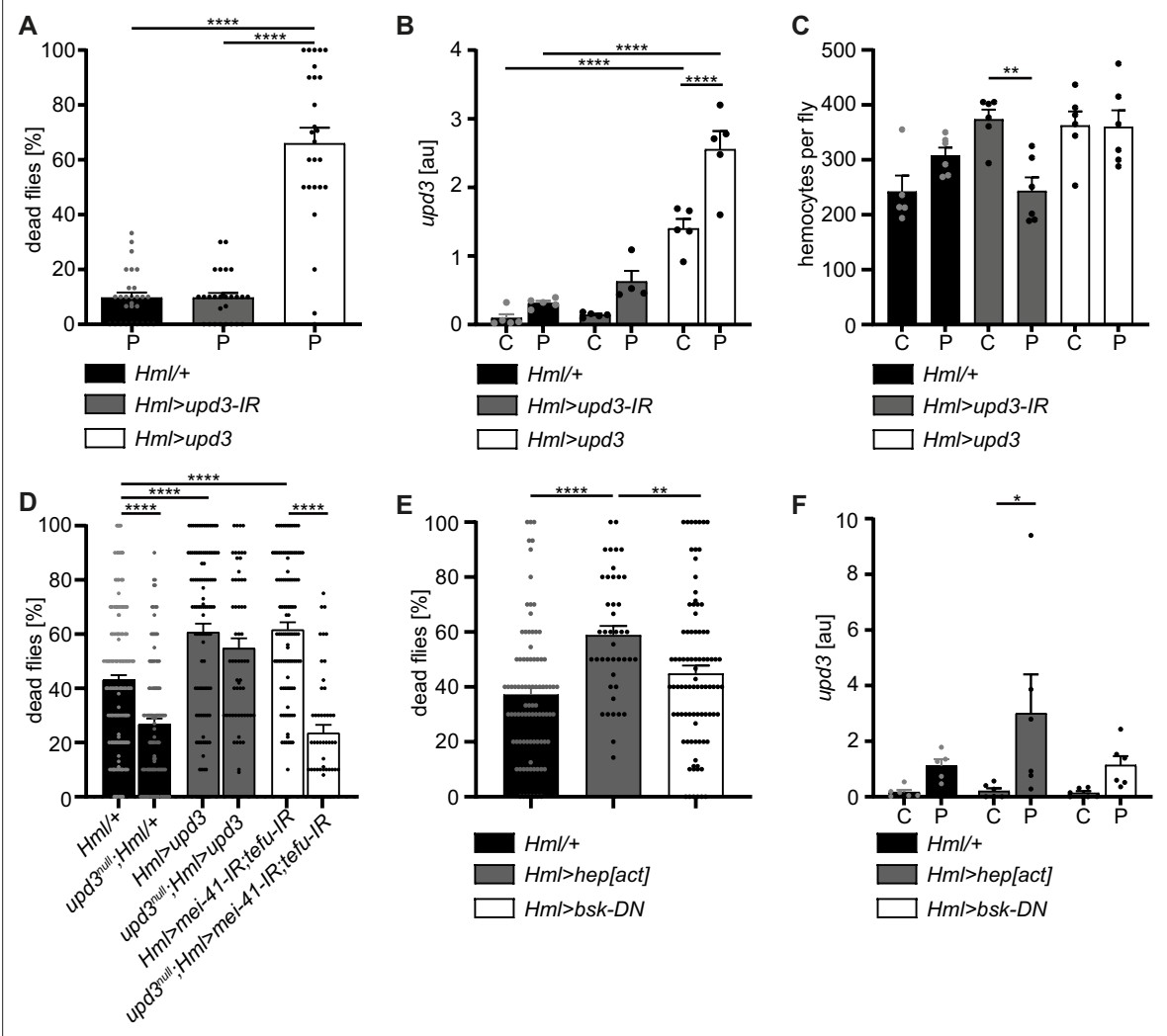

**Figure 6.** Hemocyte-derived *upd3* controls susceptibility to oxidative stress in adult *Drosophila*. (**A**) Survival of *Hml/+* (n=28), *Hml>upd3*-IR (n=27) and *Hml>upd3* (n=26) flies on 15 mM PQ food. Each data point represents a sample of 10–15 flies. Mean ± SEM is shown. Three independent experiments were performed. One-way ANOVA: ****p<0.0001. (**B**) Gene expression analysis for *upd3* via RT-qPCR in *Hml/+*, *Hml>upd3*-IR and *Hml>upd3* flies on control food (C) and 15 mM Paraquat (P). Each data point (n=5–6) represents a sample of three individual flies. Mean ± SEM is shown. Two-way ANOVA: ****p<0.0001. (**C**) Hemocyte quantifications in *Hml/+*, *Hml>upd3*-IR and *Hml>upd3* flies on control food and 15 mM PQ food. Each data point represents one fly (n=5–6). Mean ± SEM is shown. One-way ANOVA: **p<0.01. (**D**) Survival of *Hml/+* (n=241), *upd3[null];Hml/+* (n=122), *Hml>upd3* (n=104), *upd3[null];Hml>upd3* (n=57), *Hml>mei-41-IR;tefu-IR* (n=103) and *upd3[null];Hml>mei-41-IR;tefu-IR* (n=50) flies on 15 mM PQ food at 29 °C. Each data point represents a sample with 10–15 flies. Mean ± SEM is shown. One-way ANOVA: ****p<0.0001. (**E**) Survival of *Hml/+* (n=78), *Hml>hep[act]* (n=38) and *Hml>bsk*-DN (n=80) flies on 15 mM PQ food. Each data point represents a sample with 10 flies. Mean ± SEM is shown. One-way ANOVA: **p<0.01; ****p<0.0001. (**F**) Gene expression analysis for *upd3* via RT-qPCR of *Hml/+*, *Hml>hep[act]* and *Hml>bsk*-DN flies on control food (-) and 15 mM Paraquat (+). Each data point (n=5–6) represents a sample of three individual flies. Mean ± SEM is shown. Two-way ANOVA: *p<0.05.

another cellular source after eliminating hemocyte-derived *upd3*, for example fat body cells, as seen in our snRNA-seq analysis.

To identify if the higher susceptibility of *Hml>upd3* flies to oxidative stress was due to reduced hemocyte numbers, we quantified hemocytes across all three genotypes, but did not see any significant changes of hemocyte number in both genotypes compared to *Hml/+* control flies on control food and PQ (*Figure 6C* and *Figure 5—figure supplement 1F*). Only in the *Hml>upd3*-IR flies we found a reduction of hemocytes between flies on control food and PQ, but without consequences for their susceptibility to oxidative stress (*Figure 6C* and *Figure 5—figure supplement 1G*). Our data indicate that elevation of hemocyte-derived *upd3* is sufficient to increase the mortality of adult flies on PQ-mediated oxidative stress. Yet we could not exclude other cellular sources than hemocytes for *upd3* upon

oxidative stress. Therefore, we wanted to probe if hemocyte-derived *upd3* is sufficient to increase the susceptibility to PQ treatment in the absence of other cellular sources. We backcrossed *Hml>upd3* flies to *upd3*-null mutants (*upd3^null^*) to generate *upd3^null^;Hml>upd3* flies (*upd3^null^;Hml-Gal4,UAS-2xeGFP/ UAS-upd3*) and compared their survival on PQ to *Hml/+* and *upd3^null^* flies (**Figure 6D**). Of note, we found that *upd3^null^* mutants had a significant reduced susceptibility to oxidative stress compared to *Hml/+* flies, however we found a decreased survival in *upd3^null^;Hml>upd3* flies when hemocytes alone overexpressing *upd3* similar to the higher susceptibility of *Hml>upd3* flies (**Figure 6D**). We then wanted to test if loss of *upd3* would also rescue the higher susceptibility of flies with DDR-deficient hemocytes and backcrossed *Hml>mei41-IR;tefu-IR* to *upd3^null^* flies to generate *upd3^null^;Hml>mei41-IR;tefu-IR* flies (*upd3^null^;Hml-Gal4,UAS-2xeGFP/UAS-mei41-IR;UAS-tefu-IR*) (**Figure 6D**). Indeed, we found a substantially decreased susceptibility of these flies to oxidative stress compared to *Hml>mei41-IR;tefu-IR* flies, supporting our hypothesis that loss of DNA damage signaling in hemocytes and subsequently increased levels of upd3 render adult flies more susceptible to oxidative stress.

Previous studies highlighted that JNK signaling can induce *upd3* expression in hemocytes (**Woodcock et al., 2015**). In line with these reports, we found enrichment of genes with *kayak/AP-1* binding motifs in the plasmatocyte subset C6 which showed further upregulation of Jak/STAT target genes and also *upd3* expression (**Figure 3**). Therefore, we wanted to verify if JNK activation in hemocytes acts as a direct downstream signal activated by oxidative stress and subsequently induces the release of the pro-inflammatory cytokine *upd3*. We either overexpressed a constitutive active form of *JNKK hemipterous* (*hep*) in hemocytes using *Hml>hep[act]* (*w;Hml-Gal4,UAS-2xeGFP/UAS-hep[act]*) flies or overexpressed a dominant-negative version of the Jun kinase *basket* (*bsk*) in hemocytes using *Hml>bsk*-DN (*w;Hml-Gal4,UAS-2xeGFP/UAS-bsk-DN*). We compared their survival on PQ after 18 hr and in line with our previous results, we found an increased susceptibility of *Hml>hep[act]* flies to PQ treatment, but not of *Hml>bsk*-DN flies compared to the control genotype (**Figure 6E**). We again analyzed the overall lifespan of these flies on normal food and could not find differences between the genotypes (**Figure 5—figure supplement 1H**). Furthermore, when we analyzed *upd3* expression in these flies upon PQ treatment, we found a significant induction of *upd3* expression after PQ treatment

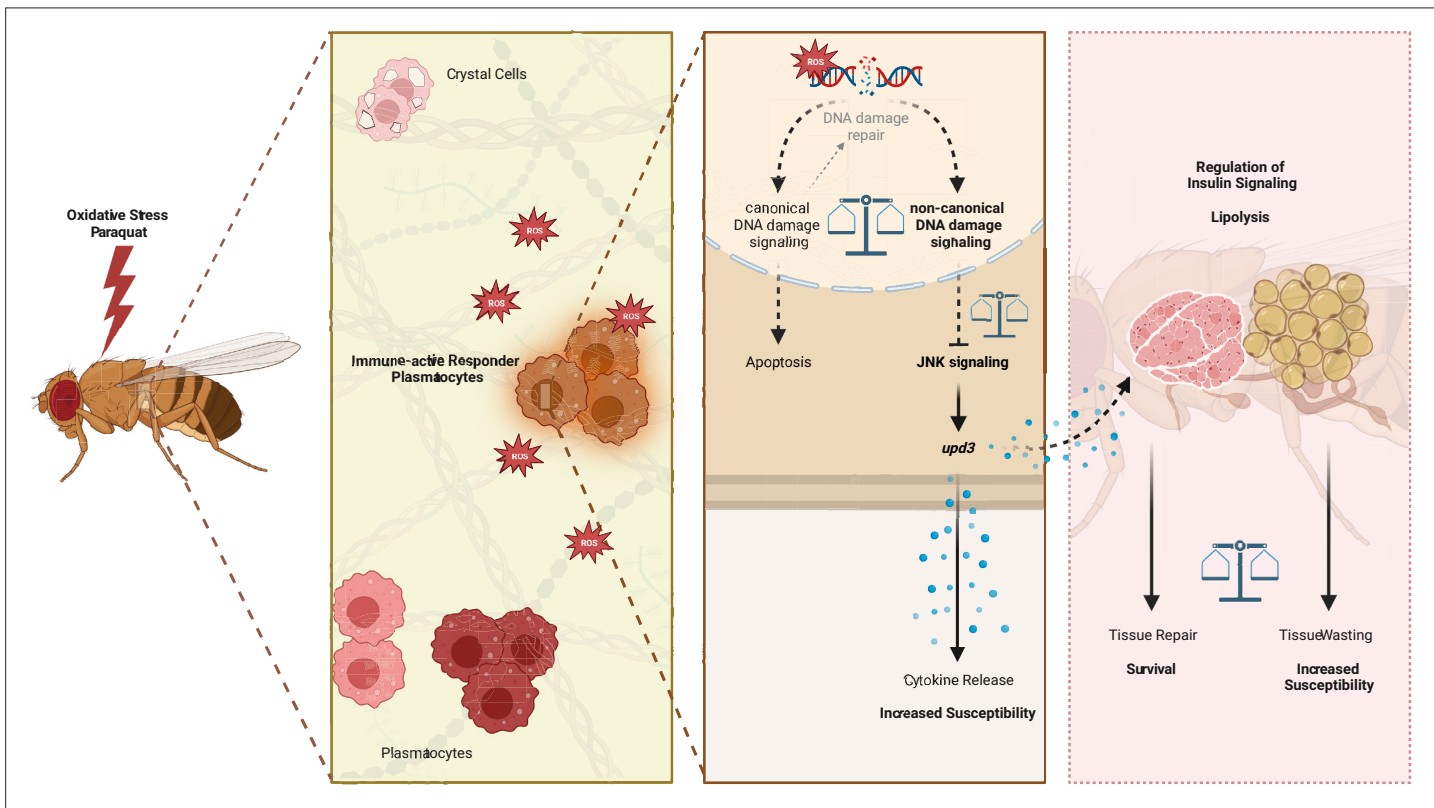

**Figure 7.** Graphical summary of the results.

in *Hml>hep[act]* compared to control food, but only a slight increase in *Hml/+* and *Hml>bsk*-DN flies upon PQ feeding (*Figure 6F*). These findings support a substantial role of JNK-mediated induction of *upd3* in hemocytes upon oxidative stress.

In summary, our study revealed a defined immune-responsive subset of plasmatocytes during oxidative stress. We show that this oxidative stress-mediated immune activation seems to be regulated by DNA damage signaling in hemocytes, which controls JNK activation and subsequent *upd3* expression (*Figure 7*). Loss of DNA damage signaling and over-induction of JNK signaling or upd3 in hemocytes release renders the adult fly more susceptible to oxidative stress (*Figure 7*).

## Discussion

Similar to vertebrate macrophages, oxidative stress triggers immune activation in *Drosophila* hemocytes (*Chakrabarti and Visweswariah, 2020*). However, this immune activation was not obvious on a systemic level upon PQ-mediated oxidative stress. In contrast, localized oxidative stress-induced immune activation in damaged tissues is an essential step of the *Drosophila* immune response to infection or injury (*Wu et al., 2012*; *Louradour et al., 2017*). However, the precise contribution of *Drosophila* macrophages in this immune activation, in the organ-to-organ communication during stress response and the implication in the organism's survival upon increased oxidative stress remained so far ill-defined. Overall, we describe the induction of diverse cellular states of adult hemocytes, namely plasmatocytes, during oxidative stress with a particular subset of plasmatocytes (cluster C6) which directly responds to oxidative stress by immune activation. We demonstrated that this immune activation seems to be at least partially controlled by the DNA damage signaling machinery in hemocytes. Loss of *nbs*, *tefu* or *mei-41* in hemocytes renders the organism more susceptible to oxidative stress without significantly affecting oxidative stress mediated DNA damage, but with an enhanced induction of JAK/STAT and JNK signaling, accompanied by changes in genes associated with energy mobilization and storage. In line with that, we describe substantial changes in the fat body on the transcriptomic level with diverse cell states, indicating decrease of triglyceride storage and induced response to oxidative stress and immune activation, but loss of AMP-producing cells. Finally, we showed that the control of hemocyte-derived *upd3* is key for the survival of the organism upon oxidative stress and that induction of the *JNK/upd3* signaling axis in hemocytes is sufficient to increase the susceptibility to oxidative stress. Hence, we postulate that hemocyte-derived *upd3* is most likely released by the activated plasmatocyte cluster C6 during oxidative stress *in vivo* and is subsequently controlling energy mobilization and subsequent tissue wasting upon oxidative stress.

*Drosophila* hemocytes have been described before to react to PQ- or infection-mediated intestinal damage via release of cytokines to promote ISC proliferation in the gut (*Ayyaz et al., 2015*; *Chakrabarti et al., 2016*). PQ is a widely used model in *Drosophila* to induce oxidative stress via oral uptake (*Rzezniczak et al., 2011*). In many reports, the gut has been described to be the most affected organ and ISC proliferation is an essential step to regenerate the injured intestinal epithelium (*Jiang et al., 2011*; *Patel et al., 2019*). PQ treatment has been described to cause gut leakage and therefore this could be a potential cause of death for the fly (*Ayyaz et al., 2015*), although we did not find signs of gut leakage within the 18 hr PQ treatment with a concentration of 15 mM. Aside from the gut, oral PQ treatment was shown to induce oxidative stress in many organs including hemocytes (*Azad et al., 2011*), oenocytes (*Huang et al., 2019*), and the brain (*Maitra et al., 2019*), indicating that the induced oxidative stress is not only localized to the gut tissue. PQ-induced oxidative stress was described before to cause neurodegeneration with parkinsonism-like symptoms, accompanied by strong immune activation and upregulation of JNK signaling in the CNS (*Maitra et al., 2019*), an activation profile similar to the identified activated plasmatocyte cluster C6 and the activated fat body clusters C2 and C4 in our snRNA-seq analysis. The snRNA-seq analysis further revealed distinct hemocyte clusters with seven plasmatocyte clusters and one crystal cell cluster. Whereas the plasmatocyte clusters C1 and C7 seem to represent cellular states found in steady state conditions only, we identified several other clusters which seemed to be exclusive or at least enriched under PQ treatment. Interestingly, we only identified one cluster, C6, which specifically responded to oxidative stress via immune activation and upregulation of oxidative stress response genes but also *Jak/STAT* and JNK target genes. We postulate that these cells might be directly exposed to the oxidative stress because their signature resembles the one of S2 cells directly treated with PQ *in vitro*. Even though oxidative DNA damage in hemocytes was followed by immune activation *in vitro* and *in vivo*, it remains to be

determined if this activation is induced by cytokines derived from damaged organs during oxidative stress or by the oxidative stress directly. The hypothesis that this cluster is directly exposed to oxidative stress is further supported by an upregulation of genes such as *Hsps* in this cluster, which are known to be directly induced by oxidative stress (*Donovan and Marr, 2016*). Furthermore, we found evidence that oxidative DNA damage and induced DNA damage signaling dynamically modulate immune activation cascades in cluster C6 plasmatocytes. This was supported by the increased DNA damage in S2 cells by PQ treatment but also the enrichment of DDR-associated genes such as *Gadd45* in cluster C6, which is associated with JNK activation and is further implicated in oxidative stress induced DNA damage (*Maitra et al., 2019*). Constant DNA damage signaling and repair is an essential step in all tissues to maintain tissue homeostasis (*Khan et al., 2019*).

In our study, we point to a potential role of DNA damage signaling as a rate-limiting step for immune activation and response to oxidative stress in hemocytes. We did not find indications of canonical DNA damage response, such as induced apoptosis in hemocytes due to the oxidative DNA damage. Non-canonical DNA damage sensing has been implicated in immune responses in vertebrates and invertebrates (*Karpac et al., 2011*; *Maitra et al., 2019*; *Dunphy et al., 2018*; *Bednarski and Sleckman, 2019*). It was demonstrated that vertebrate macrophages *in vitro* respond similarly to immune stimuli such as infection by synergistic activation of DDR and type I interferon signaling to induce downstream immune signaling cascades (*Morales et al., 2017*). Here, we provide new evidence in *Drosophila* that potentially non-canonical DNA damage is used by immune cells *in vivo* to modulate and especially control their immune response including pro-inflammatory cytokine release upon oxidative stress. Localized DNA damage in larval epithelial cells has been shown to induce non-canonical DNA damage activity leading to the secretion of *upds* and the regulation of hemocyte expansion and activation, as well as subsequent hemocyte-mediated regulation of energy storage in the fat body (*Karpac et al., 2011*). In further support of our results, one report previously demonstrated that loss of DNA damage signaling machinery induced the release of *upd3* in hemocytes (*Ayyaz et al., 2015*). Most likely, the identified plasmatocyte cluster C6 is a central cellular source of this signal. However, we also detected *upd3* induction in the PQ-induced fat body cell cluster C4 as potential secondary source of upd3, which needs to be further determined in future studies. Furthermore, anatomical localization of this upd3-producing plasmatocyte cluster, as well as spatial distribution and cross-talk of the different plasmatocyte and fat body cell clusters would be of interest. We describe a strong connection between the control of *upd3* expression by hemocytes and the susceptibility of the adult fly to oxidative stress. Upon oxidative stress, we observe decreased triglycerides and glycogen stores and altered gene expression such as induced expression of lipolytic enzymes and other *foxo* target genes indicating energy mobilization from the adult fat body, which was further supported by the transcriptomic profiling of fat body cells. Accumulating evidence throughout our study points to an implication of hemocyte-derived *upd3* in the control of energy mobilization and tissue wasting during oxidative stress. Our data from hemocyte-deficient flies indicating defects in energy mobilization and maintenance of free glucose levels further support this assumption. Our presented findings are in line with our previous studies where we showed that upon high-fat diet hemocytes produce high levels of *upd3* which induces insulin insensitivity in many tissues by activating Jak/STAT signaling in these organs and inhibiting the insulin signaling pathway (*Woodcock et al., 2015*). Furthermore, we and others described a central role for hemocyte-derived *upds* in the control of glucose metabolism in muscles during steady state and upon infection (*Kierdorf et al., 2020*; *Yang et al., 2015*). As shown before, hemocytes produce high levels of *upd3* upon infection with *Mycobacterium marinum* which also caused tissue wasting and extensive energy mobilization in a *foxo*-dependent manner (*Péan et al., 2017*; *Dionne et al., 2006*).

Our study provides further evidence that hemocytes serve as essential signaling hubs in 'organ-to-organ communication' during oxidative stress, which is in line with previous studies in immune challenge as well as steady state (*Wu et al., 2012*; *Yang et al., 2015*; *Yang and Hultmark, 2016*). Our findings in *Drosophila* highlight the role of macrophages for the survival on oxidative stress with an urgent need for a balanced immune response to overcome the tissue damage (*Figure 7*). We further unravel the diverse responses of hemocytes and fat body cells toward oxidative stress which can now serve as a baseline to further explore the spatiotemporal macrophage/hemocyte responses to oxidative stress and their communication with other host tissue cells such as the fat body (*Figure 7*). Future in-depth analysis of the regulation of these cellular states and their coupling to tissue repair will allow

to gain new insights for the treatment of diseases where tissue repair and tissue wasting needs to be balanced, including cancer or neurodegenerative diseases.

## Methods

### *Drosophila melanogaster* stocks

Flies were reared on a high yeast food containing 10% brewer's yeast, 8% fructose, 2% polenta and 0.8% Agar. Propionic acid and nipagin were added to prevent bacterial or fungal growth. All crosses (except those containing the *tub-Gal80^ts* construct or otherwise noted) were performed at 25 °C with a 12 hr dark/light cycle. The crosses with *tub-Gal80^ts* were performed at 18 °C to ensure the inhibition of the Gal4-protein during developmental stages of the fly. The experimental male F1 flies were transferred to 29 °C as soon as they hatched and were aged for six days. All transgenic lines used in this study are listed in the *Supplementary file 3*.

### Lifespan/survival assays

Male flies were collected after eclosion and groups of 20 age-matched flies per genotype were housed in a food vial. The survival experiments were performed at 29 °C. The vials were placed horizontal to avoid that flies fall into the food and become stuck. Dead flies were counted in a daily manner. The flies were transferred into a fresh food vial twice per week without $CO_2$ anesthesia.

### Paraquat treatment

Flies were maintained at 29 °C for 6 days prior to treatment. On day 6 they were starved for 6 hr. A filter paper soaked in 5% sucrose solution with or without 15 mM Paraquat (PQ, Methyl viologen hydrate, Acros Organics) was added after starvation. Flies were fed for 18 hr on the PQ containing food in groups of ten. The PQ treatment was performed at 29 °C in the dark (*Wang et al., 2003*). To assess the survival rate, the dead flies per vial were counted and the percentage was calculated. The living flies were further analyzed in other assays used in this study.

### Starvation experiments

Ten to 20 age-matched male flies were kept in a vial containing 1% agar supplemented with 2% 1xPBS. The starvation experiments were performed at 25 °C. Dead flies were counted every hour. Several individual experiments were performed and started on different daytimes to exclude diurnal derived artifacts. The data of the individual experiments were pooled and analyzed with GraphPad Prism 9.2.0.

### Paraffin sections and stainings

Anaesthetized flies were washed in 75% ethanol for 5–10 s and transferred into 4% PFA for 30 min at room temperature. The flies were washed for 1 hr in PBS on a shaker. Subsequently the flies were transferred into tissue cassettes, dehydrated, and embedded in paraffin. The flies were cut in 7-μm-thick sections de-paraffinated and stained on slides with Hematoxylin-Gill (II) for 5 min. Subsequently, the sections were blued in running water for 10 min. It followed a 0.5% Eosin treatment for 5 min and a final ethanol treatment with increasing concentrations until 100%. The sections were transferred into Xylol and finally covered with synthetic resin and a cover slip.

### Cryo sections and Oil Red O staining

Anaesthetized flies were washed in 75% ethanol for 5–10 s and transferred into 4% PFA for 30 min at room temperature. Subsequently, the flies were dehydrated for 1 hr at 37 °C in 30% sucrose solution. The dehydrated flies were placed in a Tissue-Tek cryo-mold and embedded in Tissue-Tek O.C.T. Compound. The cryo-mold was placed on dry-ice which was embedded in a 100% ethanol bath until the whole block was completely frozen. The flies were cut in 10 μm sections and stained on the slide. The slides were put on RT and dried. Subsequently, they were submerged in Oil Red O solution for 30 min. The slides were rinsed two times in deionized water and subsequently stained for 2 min with hematoxylin to stain the nuclei. Finally, the slides were washed with water for 5 min and covered with a cover slip and glycerin.

## Semi-quantitative real-time PCR

Three flies were pooled and smashed in 100 µl TRIzol to stabilize and isolate the RNA. Chloroform was used to extract the RNA followed by an isopropanol precipitation step. The RNA was cleaned with 70% ethanol and solubilized in water. A DNAse treatment was performed to digest potential genomic DNA contaminations. The purified and DNAse treated RNA was written into cDNA using RevertAid Reverse Transcriptase (Thermo Fisher Scientific) at 37 °C for 1 hr. The reaction was stopped by incubating for 10 min at 70 °C. The subsequent RTqPCR was done in SensiMix SYBR Green no-ROX (Meridian Bioscience) and was performed on a LightCycler 480 (Roche). The qPCR cycling started with a 10 min 95 °C step followed by 50 cycles with the following times and temperatures: 15 s at 95 °C, 15 s at 60 °C, and 15 s at 72 °C. The gene expression levels were normalized to the value of the measured loading control gene Rpl1.

## Confocal microscopy

The flies were anesthetized with CO2, glued on a cover slip and imaged immediately. Confocal microscopy was done with a Leica SP8 microscope (Leica) and a 10 x/NA0.4 Leica air objective. The images were acquired in a resolution of 512x512 with a scan speed of 600 Hz. Z-Stacks with a step size of 5 µm and tile scans (2x2 or 2x3 images per fly) were acquired to quantify the hemocytes in whole flies. The tiled images were merged with the LAS-X software (Leica) and maximum projections were created using Fiji/ImageJ. The hemocytes were counted in the maximum projection images of whole flies.

## Smurf assay

Smurf assays were performed to check the feeding behavior as well as the gut integrity of 7 days old, male flies upon PQ treatment. Brilliant Blue dye (FCF) was added into 5% sucrose solution in a concentration of 1% (w/v). PQ was added in concentrations of 2 mM, 15 mM and 30 mM. Control food was 5% sucrose and 1% Brilliant Blue without PQ. The flies were fed with blue food according to the PQ-treatment protocol, as described above. Flies were analyzed after 18 hr of feeding on brilliant-blue food. Three kinds of flies were distinguished. Flies that did not eat at all were excluded. Flies with a blue gut or crop were classified as 'non-smurf'. Flies which turned completely blue or showed distribution of blue dye outside the gut were classified as 'smurf'.

## Thin layer chromatography (TLC)

Each sample in the TLC analysis contains 10 flies of the respective genotype. The flies were starved for 1 hr before they were anesthetized with $CO_2$ and transferred into 100 µl chloroform:methanol (3:1) on ice. The flies were centrifuged for 5 min at 15,000 g at 4 °C and subsequently homogenized with a pestle. The sample was centrifuged again with the same settings. Lard was dissolved in chloroform:methanol (3:1) and served as triglyceride control (standard 1). Standard 1 was used to produce a standard curve with decreasing triglyceride concentrations. The samples and the standard curve were loaded on a glass silica gel plate (Millipore). The mobile phase in the TLC chamber was made with hexane:diethylether (4:1). The plate was placed in the chamber until the solvent front reached the upper end of the plate. The plate was taken out, air dried and stained with ceric ammonium heptamolybdate (CAM). Subsequently the plate was incubated at 80 °C for 2 hr to visualize the stained triglyceride bands. Images were taken with a gel documentation system (gelONE). Triglyceride density was measured and the concentrations were calculated according the pre-determined concentrations of the standard curve. Image analysis and calculations were done using ImageJ.

## Glucose assay

Seven days old male flies were used for the analysis. They were treated for 18 hr with PQ or sucrose respectively. Subsequently, the flies were starved for one hour to bring all flies into the same metabolic state. Three flies were pooled as one sample and manually homogenized in 75 µl TE +0.1% Triton X-100 (Sigma Aldrich). The samples were incubated for 20 min at 75 °C to inactivate any intrinsic enzymatic activity. 5 µl of the samples were loaded into a flat-bottom 96-well plate. Each sample was measured four times to measure glucose, trehalose, glycogen and the fly background, respectively. The fly background was determined by diluting the fly sample with water. Glucose reagent (Sentinel Diagnostics) was added to the fly sample to measure the glucose levels. Trehalase (Sigma-Aldrich) or amyloglucosidase (Sigma-Aldrich) was added to the glucose reagent to measure trehalose or

glycogen, respectively. Plates were incubated for 24 hr at 37 °C before the colorimetric measurement was performed at a wavelength of 492 nm on a Tecan Spark plate reader. The concentrations were calculated according to standards loaded on the same plate.

## Nuclei FACS sorting

A total of 100 male *Hml-dsRed.nuc* flies per sample were homogenized in 500 µl ice-cold EZ nuclei lysis buffer (Sigma-Aldrich). Another 500 µl EZ lysis buffer was added and the nuclei were incubated for 5 min on ice. Following a pre-filtration step with a 70-µm cell strainer (Miltenyi filters) the nuclei were centrifuged at 500 *g* for 6 min at 4 °C. The pellet was resuspended and incubated in 1 ml of EZ lysis buffer for 5 min on ice following a second centrifugation step with previous used settings. After discarding the supernatant, 500 µl of nuclei wash and resuspension buffer (1 x PBS; 1% BSA; 0.2 U/µl RNase inhibitor, NEB) was added without resuspending the pellet and incubated for 5 min on ice. Subsequently, the pellet was resuspended with additional 500 µl nuclei wash and resuspension buffer. The nuclei were centrifuged using previous settings and subsequently washed with 1 ml of nuclei wash and resuspension buffer. After centrifugation, the pellet was resuspended and incubated for 15 min with nuclei staining solution containing DAPI (1:1000) and DRAQ7 (1:100). The nuclei were washed once with 1 ml nuclei wash and resuspension buffer. After centrifugation, the pellet was dissolved in 200 µl wash and resuspension buffer, filtered through a 40-µm filter and sorted on a BD FACS ARIAIII. A duplet exclusion was performed by gating on singlets in FSC-A vs. FSC-W plot. Subsequently DRAQ7 +DAPI + events were sorted into nuclei wash and resuspension buffer.

## Single nucleus library preparation and sequencing

For single nucleus library preparation, nuclei were loaded onto a Chromium Single Cell 3' G Chip (10 X Genomics) to generate single-nucleus gel beads in emulsion (GEM) according to the manufacturer's instructions without modifications using Chromium Next GEM Single Cell 3' Reagents Kit v3.1 (10X Genomics). In order to multiplex the samples for sequencing, Single Index Kit T Set A (10 x Genomics) was used for library preparation according to the manufacturer's instructions. The cDNA content and size of post-sample index PCR samples was analyzed using a 2100 BioAnalyzer (Agilent). Library quantification was done using NEB Next Library Quant Kit for Illumina (New England Biolabs) following manufacturer's instructions. Sequencing libraries were loaded on an Illumina Nextseq 550 flow cell, with sequencing settings according to the recommendations of 10×Genomics. Sample de-multiplexing was done using built-in BCL2FASTQ. Cell Ranger v6 software was implemented for gene alignment to the fly genome. The *Drosophila melanogaster* genome assembly and annotation file were downloaded from Ensembl (Release 104). The annotation files were filtered as suggested by 10 x Genomics, to keep only the categories of interest (i.e. protein coding genes, long intergenic noncoding RNAs, antisense, pseudogenes). Cell Ranger v6 with the 'include-intron' parameter was used for the generation of the count data.

## snRNA-seq data analysis

Downstream analysis implemented the SoupX v1 and Seurat v4 R-based packages. The matrix for each sample was loaded using the SoupX package (*Young and Behjati, 2020*) and ambient RNA contamination was calculated using the default setting. The corrected matrix together with metadata on treatment were used to create an object in Seurat v4 (*Hao et al., 2021*). Doublet detection was done using the DoubletFinder package v2 (*McGinnis et al., 2019*). After doublet exclusion, a combined object was created from the list by using the 'merge' function. Nuclei with <5% of mitochondrial contamination and between 400 and 1700 genes expressed were retained for further analysis. The filtered raw count matrix was log-normalized within each nucleus. The top 3000 variable genes were calculated by Seurat using the variance stabilizing transformation selection method and data were scaled. The variable genes were used to perform principal component analysis (PCA) and the top 20 principal components were used for the unsupervised clustering. Seurat applies a graph-based clustering approach: the 'FindNeighbors' function uses the number of principal components (dims = 20) to construct the k-nearest neighbors graph based on the Euclidean distance in PCA space, and the 'FindClusters' function applies the Louvain algorithm to iteratively group nuclei (resolution = 1). We then used the PCA embedding to generate an UMAP for cluster visualization. For each cluster, differentially expressed genes (DEGs) were calculated using the 'FindMarkers' and 'FindAllMarkers' function with

default parameters with adjusted p < 0.05 using Bonferroni correction. The DEGs were used to assign cell type identity to clusters based on known cell lineage markers described in previously published study (*Cattenoz et al., 2020*). To understand the impact of Paraquat treatment on each cluster, 'Find-Markers' function was used to compare 'control' vs 'Paraquat' treated cells per cluster. Based on the DEGs obtained from this analysis, volcano plots were created using the EnhancedVolcano package v1 (*Blighe, 2021*). UMAP visualizations, feature plots and dot plots for snRNA-seq data were generated using in-built plotting functionality of Seurat.

## Subclustering of hemocytes and further analysis

Based on the cell identity and known hemocyte marker genes from a previous study (*Cattenoz et al., 2020*), sub-clusterings were performed using the same unsupervised algorithm and downstream analysis. After removing non-hemocyte clusters, final sub-clustering was done with a modification that 'FindNeighbors' function was run with dims = 10 and 'FindClusters' function was run with resolution 0.8. Based on the DEGs found by the 'FindAllMarkers' function (parameters: only.pos=T, test.use = 'MAST') a heat map was plotted for TOP 20 DE genes per clusters.

## Subclustering of fat body cells and further analysis

Based on the cell identity according to the FlyCell Atlas, sub-clustering was performed using the same unsupervised algorithm and downstream analysis as used for hemocytes. After removing non-fat body clusters, final sub-clustering was done with the 'FindNeighbors' function which was run with dims = 10 and 'FindClusters' function which was run with resolution 0.8. Based on the DEGs found by the 'FindAllMarkers' function (parameters: only.pos=T, test.use = "MAST"), a heat map was plotted for TOP 20 DE genes per clusters.

## Regulation network analysis of hemocyte clusters

In order to determine the transcription factor activity, regulon analysis was performed using the R package SCENIC (v1.3.1) (*Aibar et al., 2017*). To remove noise, genes with low expression levels or low positive rates were filtered using the 'geneFiltering' function with default settings to obtain a filtered matrix which was used to build co-expression network using the 'runCorrelation' and 'runGENIE3' functions. Gene regulatory networks (GRNs) were built and scored using the default parameters. Potential regulons based on DNA-motif analysis were selected by using RcisTarget and active gene networks were identified by AUCell. Regulon activity for each cell was calculated as the average normalized expression of putative target genes. Regulon activity matrix was exported which was used to create a new 'AUC' assay using 'CreateAssayObject' function of Seurat. 'DoHeatmap' function from Seurat was used to plot scaled TF activity in individual hemocytes.

## S2 cell culture, CellROX staining, and flow cytometry

*Drosophila* S2 cells (Thermo Scientific, R96007) were cultured in Schneider's *Drosophila* medium (Gibco) at 26 °C and atmospheric oxygen and carbon dioxide conditions. The medium contained 10% FCS and 1% penicillin/streptomycin. PQ was added into the medium in concentrations of 15 mM or 30 mM to investigate the influence on the production of reactive oxygen species (ROS). Control cells remained untreated. To visualize ROS and oxidative stress, the cells were stained with CellROX Deep Red Reagent (Invitrogen) for 30 min directly in the medium in a concentration of 1:500 at 26 °C. The cells were washed twice with 1xPBS containing 2 mM EDTA. In order to stain dead cells, DAPI was added to the samples in a concentration of 1:1000. Samples were acquired using a BD LSRFortessa (BD Biosciences) and analyzed with FlowJo analysis software.

## COMET assays

Comet assays were performed to analyze the amount of DNA damage in S2 cells treated with PQ. S2 cells were treated with 15 mM or 30 mM PQ for 24 hr. The cells were brought into a concentration of $1.8 \times 10^5$ cells/ml. Low melting agarose (LMA) was boiled at 90 °C and subsequently cooled down to 37 °C. The cells were diluted in low melting agarose (LMA) at a ratio of 1:10 (cells:LMA) and transferred onto a CometSlide (Trevigen). To ensure proper attachment of the LMA, the slides were cooled for 30 min at 4 °C. The lysis was performed for 1 hr at 4 °C. Therefore, the slides were submerged in CometAssay Lysis Solution (Trevigen). Subsequently, the cells were submerged in alkaline unwinding

solution (200 mM NaOH, 1 mM EDTA, pH 13) for 1 hr at 4 °C. The electrophoresis was performed under alkaline conditions (pH13) for 30 min with a current of 20 V (1 V/cm). The slides were neutralized with ddH$_2$O, dehydrated with 37% ethanol and dried properly at 37 °C before they were stained with SYBRGold (Invitrogen) for 30 min. Comets were imaged with an Olympus BX61 fluorescence microscope. Comet data was analyzed via TriTrek CometScore 2.0.0.38 software.

### Statistical analysis and data handling

Statistical significance of real-time qPCR, Paraquat survival, hemocyte quantification, TLC, Glucose assay, CellROX and comet assay data was calculated with an unpaired t-test or one-way ANOVA as indicated in the figure legends. Lifespan/survival assays were analyzed using Log-Rank and Wilcoxon test. Significance digits indicate the following significance levels: *p<0.05, **p<0.01, ***p<0.001, ****p<0.0001. Significance tests were performed using GraphPad Prism 9 software. Data handling of snRNAseq data is described above.

## Acknowledgements

The authors are very grateful to the members of the Kierdorf lab for their help and support. The authors would especially like to thank M Oberle for her exceptional help with the maintenance of the *Drosophila* stocks and the management of the fly room. We would like to acknowledge the Lighthouse Core Facility and its staff, especially J Bodinek-Wersing and U Jagadeshwaran, for their assistance with the sorting of nuclei. The authors are especially grateful to D Ganser and H Fischer for the continuous support throughout the study. Schemes were created with BioRender.com.

## Additional information

### Funding

| Funder | Grant reference number | Author |
|---|---|---|
| Deutsche Forschungsgemeinschaft | 259373024 | Marco Prinz |
| Fritz Thyssen Stiftung | Project Grant | Katrin Kierdorf |
| Deutsche Forschungsgemeinschaft | 441891347 | Marco Prinz<br>Katrin Kierdorf<br>Olaf Groß |
| Deutsche Forschungsgemeinschaft | 390939984 | Anne-Kathrin Classen<br>Katrin Kierdorf<br>Olaf Groß<br>Marco Prinz |
| Boehringer Ingelheim Foundation | Plus 3 Programme | Anne-Kathrin Classen |
| European Research Council | 337689 | Olaf Groß |
| European Research Council | 966687 | Olaf Groß |
| Deutsche Forschungsgemeinschaft | KI 1876/3-1 | Katrin Kierdorf |
| Deutsche Forschungsgemeinschaft | 423813989 | Olaf Groß |
| Deutsche Forschungsgemeinschaft | 422681845 | Olaf Groß |
| Deutsche Forschungsgemeinschaft | 256073931 | Marco Prinz<br>Katrin Kierdorf<br>Olaf Groß |

| Funder | Grant reference number | Author |
|---|---|---|
| European Research Council | 101034170 | Olaf Groß |
| Deutsche Forschungsgemeinschaft | 192904750 | Marco Prinz |
| Novo Nordisk Fonden | Novo Nordisk Award | Marco Prinz |
| Jung-Stiftung für Wissenschaft und Forschung | Ernst Jung Medal | Marco Prinz |
| Deutsche Forschungsgemeinschaft | Reinhart Koselleck Grant | Marco Prinz |
| Deutsche Forschungsgemeinschaft | Gottfried Wilhelm Leibniz Prize | Marco Prinz |
| Ministerium für Wissenschaft, Forschung und Kunst Baden-Württemberg | Sonderlinie "Neuroinflammation" | Marco Prinz |
| Alzheimer Forschung Initiative | | Marco Prinz |
| Wellcome Trust | 10.35802/207467 | Marc S Dionne |
| Biotechnology and Biological Sciences Research Council | BB/W/001004/1 | Marc S Dionne |
| European Research Council | Marie Skłodowska-Curie Postdoctoral Fellowship | Gianni Monaco |

The funders had no role in study design, data collection and interpretation, or the decision to submit the work for publication. For the purpose of Open Access, the authors have applied a CC BY public copyright license to any Author Accepted Manuscript version arising from this submission.

## Author contributions

Fabian Hersperger, Data curation, Formal analysis, Validation, Investigation, Visualization, Methodology, Writing – original draft, Writing – review and editing; Tim Meyring, Data curation, Formal analysis, Visualization; Pia Weber, Data curation, Formal analysis; Chintan Chhatbar, Data curation, Software, Formal analysis, Methodology; Gianni Monaco, Software, Formal analysis, Methodology; Marc S Dionne, Katrin Paeschke, Marco Prinz, Olaf Groß, Anne-Kathrin Classen, Resources, Writing – original draft, Writing – review and editing; Katrin Kierdorf, Conceptualization, Formal analysis, Supervision, Funding acquisition, Investigation, Writing – original draft, Project administration, Writing – review and editing

## Author ORCIDs

Marc S Dionne ⓘ https://orcid.org/0000-0002-8283-1750
Katrin Paeschke ⓘ https://orcid.org/0000-0003-3080-6745
Anne-Kathrin Classen ⓘ https://orcid.org/0000-0001-5157-0749
Katrin Kierdorf ⓘ https://orcid.org/0000-0002-9272-4780

Reviewer #1 (Public Review): https://doi.org/10.7554/eLife.86700.3.sa1
Reviewer #2 (Public Review): https://doi.org/10.7554/eLife.86700.3.sa2
Reviewer #3 (Public Review): https://doi.org/10.7554/eLife.86700.3.sa3
Author response https://doi.org/10.7554/eLife.86700.3.sa4

# Additional files

## Supplementary files

• Supplementary file 1. Differentially expressed genes in hemocyte clusters C1-C8.

- Supplementary file 2. Differentially expressed genes in fat body cell clusters C1-C8.
- Supplementary file 3. Transgenic *Drosophila* lines and primers used in the study.
- MDAR checklist

## Data availability

snRNA-seq data is available on the gene omnibus express (GEO) platform, GEO accession number: GSE244295. Microcopy, flow cytometry, qPCR, glucose/glycogen assay, TLC and survival data reported in the paper is shared via Dryad.

The following datasets were generated:

| Author(s) | Year | Dataset title | Dataset URL | Database and Identifier |
|---|---|---|---|---|
| Kierdorf K, Hersperger F, Monaco G, Chhatbar C | 2023 | Single nuclei transcriptomic profiling of adult *Drosophila* upon oxidative stress | https://www.ncbi.nlm.nih.gov/geo/query/acc.cgi?acc=GSE244295 | NCBI Gene Expression Omnibus, GSE244295 |
| Hersperger F, Meyring T, Weber P, Chhatbar C, Monaco G, Dionne MS, Paeschke K, Prinz M, Groß O, Classen AK, Kierdorf K | 2024 | *Drosophila* macrophage response to oxidative stress | https://doi.org/10.5061/dryad.6hdr7sr78 | Dryad Digital Repository, 10.5061/dryad.6hdr7sr78 |

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
