## [Editor Report · eLife assessment]

This study elucidates the role of a specific hemocyte subpopulation in oxidative damage response by establishing connections between DNA damage response and the JNK-JAK/STAT axis to regulate energy metabolism. The identification of this distinct hemocyte subpopulation through single-cell RNA sequencing analysis and the finding of hemocytes that respond to oxidative stress are **important**. The method for single-cell RNA sequencing and related analyses are **convincing** and experiments linking oxidative stress to DNA damage and energy expenditure are **solid**. The finding of stress-responsive immune cells capable of influencing whole-body metabolism adds insights for cell biologists and developmental biologists in the fields of immunology and metabolism.

---

## [Referee Report · Reviewer #1 (Public Review)]

The study examines how hemocytes control whole-body responses to oxidative stress. Using single cell sequencing they identify several transcriptionally distinct populations of hemocytes, including one subset that show altered immune and stress gene expression. They also find that knockdown of DNA Damage Response (DDR) genes in hemocytes increases expression of the immune cytokine, upd3, and that both upd3 overexpression in hemocytes and hemocyte knockdown of DDR genes leads to increased lethality upon oxidative stress. And they find that the PQ-induced lethality seen when the DDR is disrupted can be rescued in upd3 null background, suggesting links between proper regulation of DDR in hemocytes, modulation of systemic upd3 signaling, and the control of oxidative stress survival.

The paper has two key strengths:

1, The single cell analyses provide a clear description of how oxidative stress can cause distinct transcriptional changes in different populations of hemocytes. These results add to the emerging them in the field that there functionally different subpopulations of hemocytes that can control organismal responses to stress.

2, The discovery that DDR genes are required upon oxidative stress to modulate upd3 cytokine production and lethality provides interesting new insight into the DDR may play non-canonical roles in controlling organismal responses to stress.

---

## [Referee Report · Reviewer #2 (Public Review)]

Hersperger et al. investigated the importance of *Drosophila* immune cells, called hemocytes, in the response to oxidative stress in adult flies. They found that hemocytes are essential in this response, and using state-of-the-art single-cell transcriptomics, they identified expression changes at the level of individual hemocytes. This allowed them to cluster hemocytes into subgroups with different responses, which certainly represents very valuable work. One of the clusters appears to respond directly to oxidative stress and shows a very specific expression response that could be related to the observed systemic metabolic changes and energy mobilization.

Using hemocyte-specific genetic manipulation, the authors convincingly show that the DNA damage response in hemocytes regulates JNK activity and subsequent expression of the JAK/STAT ligand Upd3. Silencing of the DNA damage response or excessive activation of JNK and Upd3 leads to increased susceptibility to oxidative stress. This nicely demonstrates the importance of tight control of JNK-Upd3 signaling in hemocytes during oxidative stress. The treatment the authors used is quite harsh, and in such a situation it is simply better not to use upd3 signaling, but it is still worth bearing in mind that upd3 signaling may have a protective role under milder stresses, but Upd3 could require very tight control - this could be an interesting objective for future studies.

The authors demonstrate that hemocytes play an important role in energy mobilization during oxidative stress, suggesting that control of energy mobilization by hemocytes is essential for the response. They further postulate that "hemocyte-derived upd3 is most likely released by the activated plasmatocyte cluster C6 during oxidative stress in vivo and is subsequently controlling energy mobilization and subsequent tissue wasting upon oxidative stress." It is important to note here that the association of upd3 with the observed changes in energy metabolism has not been tested, and the subsequent tissue wasting allegedly caused by excessive upd3 as a cause of death remains an open question.

---

## [Referee Report · Reviewer #3 (Public Review)]

In this study, Kierdorf and colleagues investigated the function of hemocytes in oxidative stress response and found that non-canonical DNA damage response (DDR) is critical for controlling JNK activity and the expression of cytokine unpaired3. Hemocyte-mediated expression of upd3 and JNK determines the susceptibility to oxidative stress and systemic energy metabolism required for animal survival, suggesting a new role for hemocytes in the direct mediation of stress response and animal survival.

In the revised manuscript, the authors provide additional evidence to support the role of DNA damage-modulated cytokine release by hemocytes during oxidative stress responses and strengthen the connection between DNA damage and the regulation of upd3 release from hemocytes. The authors have also included new analyses to emphasize the significance of hemocytes in the regulation of energy during oxidative stress. Following the reviewers' suggestions, the authors made improvements to the manuscript and the graphical abstract to better display their findings. Overall, the revised manuscript makes it easier to understand the main points, flows better, and is supported by convincing data and analysis throughout.

---

## [Author Response]

The following is the authors’ response to the original reviews.

**eLife assessment**
This study reports important findings regarding the systemic function of hemocytes controlling whole-body responses to oxidative stress. The evidence in support of the requirement for hemocytes in oxidative stress responses as well as the hemocyte single-nuclei analyses in the presence or absence of oxidative stress are convincing. In contrast, the genetic and physiological analyses that link the non-canonical DDR pathway to upd3/JNK expression and high susceptibility, and the inferences regarding the function of hemocytes in systemic metabolic control are incomplete and would benefit from more rigorous approaches. The work will be of interest to cell and developmental biologists working on animal metabolism, immunity, or stress responses.

We would like to thank the editorial team for these positive comments on our manuscript and the constructive suggestions to improve our manuscript. We are now happy to send you our revised manuscript, which we improved according to the suggestions and valuable comments of the referees.

**Public Reviews:**

**Reviewer #1 (Public Review):**
The study examines how hemocytes control whole-body responses to oxidative stress. Using single cell sequencing they identify several transcriptionally distinct populations of hemocytes, including one subset that show altered immune and stress gene expression. They also find that knockdown of DNA Damage Response (DDR) genes in hemocytes increases expression of the immune cytokine, upd3, and that both upd3 overexpression in hemocytes and hemocyte knockdown of DDR genes leads to increased lethality upon oxidative stress.Strengths1. The single cell analyses provide a clear description of how oxidative stress can cause distinct transcriptional changes in different populations of hemocytes. These results add to the emerging them in the field that there functionally different subpopulations of hemocytes that can control organismal responses to stress.1. The discovery that DDR genes are required upon oxidative stress to limit cytokine production and lethality provides interesting new insight into the DDR may play non-canonical roles in controlling organismal responses to stress.

We are grateful to referee 1 to point out the importance and novelty of our snRNA-seq data and our findings on the role of DNA damage-modulated cytokine release by hemocytes during oxidative stress. We further extended these analyses in the revised manuscript by looking deeper into the transcriptomic alterations in fat body cells upon oxidative stress (Figure 4, Figure S4). We further provide additional data to support the connection of DNA damage signaling and regulation of upd3 release from hemocytes (Figure 6F). Here we show that upd3-deficiency can abrogate the increased susceptibility of flies with mei41 and tefu knockdown in hemocytes. In line with this finding, we also show that upd3null mutants show a reduced but not abolished susceptibility to oxidative stress overall (Figure 6F), underlining the role of upd3 as a mediator of oxidative stress response.

Weaknesses1. In some ways the authors interpretation of the data - as indicated, for example, in the title, summary and model figure - don't quite match their data. From the title and model figure, it seems that the authors suggest that the DDR pathway induces JNK and Upd3 and that the upd3 leads to tissue wasting. However, the data suggest that the DDR actually limits upd3 production and susceptibility to death as suggested by several results:

According to the referee’s suggestion, we revised the manuscript and adjusted our title, abstract and graphical summary to be more precise that DNA damage signaling seem to have a modulatory or regulatory effect on upd3 release. Furthermore, we provide now additional data to support the connection between DNA damage signaling and upd3 release. For example, we added several genetic “rescue” experiments to strengthen the epistasis that modulation of DNA damage signaling and the higher susceptibility of the fly is connected to altered upd3 levels (Figure 6F). We now provide additional data showing that the loss of upd3 rescues the susceptibility to oxidative stress in flies, which are deficient for DDR components in hemocytes.

a. PQ normally doesn't induce upd3 but does lead to glycogen and TAG loss, suggesting that upd3 isn't connected to the PQ-induced wasting.

Even though in our systemic gene expression analysis of upd3 expression, we could not detect a significant induction of upd3 upon PQ feeding. However, we found upd3 expression within our snRNAseq data in a distinct cluster of immune-activated hemocytes (Figure 3B, Cluster 6). Upon knockdown of the DNA damage signaling in hemocytes, the levels then increase to a detectable level in the whole fly. This supports our assumption that upd3 is needed upon oxidative stress to induce energy mobilization from the fat body, but needs to be tightly controlled to balance tissue wasting for energy mobilization. Furthermore, we found evidence in our new analysis of the snRNA-seq data of the fat body cells, that indeed we can find Jak/STAT activation in one cell cluster here, which could speak for an interaction of Cluster 6 hemocytes with cluster 6 fat body cells. A hypothesis we aim to explore in future studies.

b. knockdown of DDR upregulates upd3 and leads to increased PQ-induced death. This would suggest that activation of DDR is normally required to limit, rather than serve as the trigger for upd3 production and death.

Our data support the hypothesis that DDR signaling in hemocytes “modulates” upd3 levels upon oxidative stress. We now carefully revised the text and the graphical summary of the manuscript to emphasize that oxidative stress causes DNA damage, which subsequently induces the DNA damage signaling machinery. If this machinery is not sufficiently induced, for example by knockdown of tefu and mei-41, non-canonical DNA damage signaling is altered which induces JNK signaling and induces release of pro-inflammatory cytokines, including upd3. Whereas DNA damage itself is only slightly increase in the used DDR deficient lines (Figure 5C) and hemocytes do not undergo apoptosis (unaltered cell number on PQ (Figure 5B)), we conclude that loss of tefu, mei-41, or nbs1 causes dysregulation of inflammatory signaling cascades via non-canonical DNA damage signaling. However, oxidative stress itself seems to also induce upd3 release and DNA damage signaling in the same cell cluster, as shown by our snRNA-seq data (Figure 3B). Hence, we think that DNA damage signaling is needed as a rate-limiting step for upd3 release.

c. hemocyte knockdown of either JNK activity or upd3 doesn't affect PQ-induced death, suggesting that they don't contribute to oxidative stress-induced death. It’s only when DDR is impaired (with DDR gene knockdown) that an increase in upd3 is seen (although no experiments addressed whether JNK was activated or involved in this induction of upd3), suggesting that DDR activation prevents upd3 induction upon oxidative stress.

Whereas the double knockdown of upd3 or bsk and DDR genes was resulting in insufficient knockdown efficiencies, we added a rescue experiment where we combined upd3null mutants with knockdown of tefu and mei-41 in hemocytes and found a reduced susceptibility of DDR-deficient flies to oxidative stress.

1. The connections between DDR, JNK and upd3 aren't fully developed. The experiments show that susceptibility to oxidative stress-induced death can be caused by (a) knockdown of DDR genes, (b) genetic overexpression of upd3, (c) genetic activation of JNK. But whether these effects are all related and reflect a linear pathway requires a little more work. For example, one prediction of the proposed model is that the increased susceptibility to oxidative stress-induced death in the hemocyte DDR gene knockdowns would be suppressed (perhaps partially) by simultaneous knockdown of upd3 and/or JNK. These types of epistasis experiments would strengthen the model and the paper.

As mentioned before, we had some technical difficulties combining the knockdown of bsk or upd3 with DDR genes. However, we added a new experiment in which we show that upd3null mutation can rescue the higher susceptibility of hemocytes with tefu and mei41 knockdown.

1. The (potential) connections between DDR/JNK/UPD3 and the oxidative stress effects on depletion of nutrient (lipids and glycogen) stores was also not fully developed. However, it may be the case that, in this paper, the authors just want to speculate that the effects of hemocyteDDR/upd3 manipulation on viability upon oxidative stress involve changes in nutrient stores.

In the revised version of the manuscript, we now provide a more thorough snRNA-seq analysis in the fat body upon PQ treatment to give more insights on the changes in the fat body upon PQ treatment. We added additional histological images of the abdominal fat body on control food and PQ food, to demonstrate the elimination of triglycerides from fat body with Oil-Red-O staining (Figure S1). We also analyzed now hemocyte-deficient (crq-Gal80ts>reaper) flies for their levels of triglycerides and carbohydrates during oxidative stress, to support our hypothesis that hemocytes are key players in the regulation of energy mobilization during oxidative stress. Loss of hemocytes (and therefore also their regulatory input on energy mobilization from the fat body) results in increased triglyceride storage in the fat body during steady state with a decreased consumption of these triglycerides on PQ food compared to control flies (Figure 1J). In contrast, glycogen storage and mobilization, which is mostly done in muscle, is not altered in these flies during oxidative stress (Figure 1L). Interestingly, free glucose levels are drastically reduced in hemocyte-deficient flies, which could be due to insufficient energy mobilization from the fat body and subsequently results in a higher susceptibility of these flies on oxidative stress (Figure 1K). Additionally, we aim to point out here that “functional” hemocytes are needed for effective response to oxidative stress, but this response has to be tightly balanced (see also new graphical abstract).

**Reviewer #2 (Public Review):**
Hersperger et al. investigated the importance of *Drosophila* immune cells, called hemocytes, in the response to oxidative stress in adult flies. They found that hemocytes are essential in this response, and using state-of-the-art single-cell transcriptomics, they identified expression changes at the level of individual hemocytes. This allowed them to cluster hemocytes into subgroups with different responses, which certainly represents very valuable work. One of the clusters appears to respond directly to oxidative stress and shows a very specific expression response that could be related to the observed systemic metabolic changes and energy mobilization. However, the association of these transcriptional changes in hemocytes with metabolic changes is not well established in this work. Using hemocyte-specific genetic manipulation, the authors convincingly show that the DNA damage response in hemocytes regulates JNK activity and subsequent expression of the JAK/STAT ligand Upd3. Silencing of the DNA damage response or excessive activation of JNK and Upd3 leads to increased susceptibility to oxidative stress. This nicely demonstrates the importance of tight control of JNK-Upd3 signaling in hemocytes during oxidative stress. However, it would have been nice to show here a link to systemic metabolic changes, as the authors conclude that it is tissue wasting caused by excessive Upd3 activation that leads to increased susceptibility, but metabolic changes were not analyzed in the manipulated flies.

We thank the referee for the suggestion to better connect upd3 cytokine levels to energy mobilization from the fat body. We agree that this is an important point to support our hypothesis. First, we added now a detailed analysis of fat body cells in our snRNA-seq data to evaluate the changes induced in the fat body upon oxidative stress. We further added additional metabolic analyses of hemocyte-deficient flies (crq-Gal80ts>reaper) to support our hypothesis that hemocytes are key players in the regulation of energy mobilization during oxidative stress (see also answer to referee 1). Loss of the regulatory role of hemocytes in the energy mobilization and redistribution leads to a decreased consumption of these triglycerides on PQ food compared to control flies (Figure 1J). In contrast, glycogen storage and mobilization from muscle, is not affected in hemocyte-deficient flies during oxidative stress (Figure 1L). Interestingly, free glucose levels are drastically reduced in hemocyte-deficient flies compared to controls, which could be due to insufficient energy mobilization from the fat body resulting in a higher susceptibility to oxidative stress (Figure 1K). This data supports our assumption that “functional” hemocytes are needed for effective response to oxidative stress, but this response has to be tightly balanced (see also new graphical summary).

The overall conclusion of this work, as presented by the authors, is that Upd3 expression in hemocytes under oxidative stress leads to tissue wasting, whereas in fact it has been shown that excessive hemocyte-specific Upd3 activation leads to increased susceptibility to oxidative stress (whether due to increased tissue wasting remains a question). The DNA damage response ensures tight control of JNK-Upd3, which is important. However, what role naturally occurring Upd3 expression plays in a single hemocyte cluster during oxidative stress has not been tested. What if the energy mobilization induced by this naturally occurring Upd3 expression during oxidative stress is actually beneficial, as the authors themselves state in the abstract - for potential tissue repair? It would have been useful to clarify in the manuscript that the observed pathological effects are due to overactivation of Upd3 (an important finding), but this does not necessarily mean that the observed expression of Upd3 in one cluster of hemocytes causes the pathology.

We agree with the referee that the pathological effects and increased susceptibility to oxidative stress are mediated by over-activated hemocytes and enhanced cytokine release, including upd3 during oxidative stress. We edited the revised manuscript accordingly to imply a “regulatory” role of upd3, which we suspect and suggest as an important mediator for inter-organ communication between hemocytes and fat body. Whereas our used model for oxidative stress (15mM Paraquat feeding) is a severe insult from which most of the flies will not recover, we could not account and test how upd3 might influence tissue repair after injury, insults and infection. We believe that this is an important factor, we aim to explore in future studies.

**Reviewer #3 (Public Review):**
In this study, Kierdorf and colleagues investigated the function of hemocytes in oxidative stress response and found that non-canonical DNA damage response (DDR) is critical for controlling JNK activity and the expression of cytokine unpaired3. Hemocyte-mediated expression of upd3 and JNK determines the susceptibility to oxidative stress and systemic energy metabolism required for animal survival, suggesting a new role for hemocytes in the direct mediation of stress response and animal survival.Strength of the study:1. This study demonstrates the role of hemocytes in oxidative stress response in adults and provides novel insights into hemocytes in systemic stress response and animal homeostasis.1. The single-cell transcriptome profiling of adult hemocytes during Paraquat treatment, compared to controls, would be of broad interest to scientists in the field.

We are grateful to these positive comments on our data and are excited that the referee pointed out the importance of our provided snRNA-seq analysis of hemocytes and other cell types during oxidative stress. In the revised, version we now extended this analysis and looked not only into hemocytes but also highlighted induced changes in the fat body (Figure 4).

Weakness of the study:1. The authors claim that the non-canonical DNA damage response mechanism in hemocytes controls the susceptibility of animals through JNK and upd3 expression. However, the link between DDR-JNK/upd3 in oxidative stress response is incomplete and some of the descriptions do not match their data.

In the revised manuscript, we aimed to strengthen the weaknesses pointed out by the referee. We now included additional genetic crosses to validate the connection of DDR signaling in hemocytes with upd3 release. For example, we added now survival studies where we show that upd3null mutation can rescue the higher susceptibility of flies with tefu and mei41 knockdown in hemocytes during oxidative stress. Furthermore, we added additional data to highlight the importance of hemocytes themselves as essential regulators of susceptibility to oxidative stress. We analyzed the hemocyte-deficient flies (crq-Gal80ts>reaper) for their triglyceride content and carbohydrate levels during oxidative stress (Figure 1 I-L). As outlined above, loss of hemocytes leads to a decreased consumption of these triglycerides on PQ food compared to control flies (Figure 1J). In contrast, glycogen storage and mobilization from muscle, is not affected in hemocyte-deficient flies during oxidative stress (Figure 1L). Interestingly, free glucose levels are drastically reduced in hemocyte-deficient flies, which could be due to insufficient energy mobilization from the fat body resulting in a higher susceptibility to oxidative stress (Figure 1K).

1. The schematic diagram does not accurately represent the authors' findings and requires further modifications.

We carefully revised the text throughout the manuscript describing our results and edited the graphical abstract to display that upd3 levels and hemocytes are essential to balance and modulate response to oxidative stress.

**Reviewer #1 (Recommendations For The Authors):**
The summary doesn't say too much about what the specific discoveries and results of the study are. The description is limited to just one sentence saying, "Here we describe the responses of hemocytes in adult *Drosophila* to oxidative stress and the essential role of non-canonical DNA damage repair activity in direct "responder" hemocytes to control JNK-mediated stress signaling, systemic levels of the cytokine upd3 and subsequently susceptibility to oxidative stress" which doesn't provide sufficient explanation of what the results were.

In the revised version of our manuscript, we now provide further information for the reader to outline the findings of our study in a concise way in the summary.

**Reviewer #2 (Recommendations For The Authors):**
1. To strengthen the conclusion that the DDR response suppresses JNK, and thus Upd3, rescue of DDR by upd3 null mutation would help (knockdown by Hml>upd3IR might not work, RNAi seems problematic).

We would like to thank the referee for this suggestion and included now a genetic experiment where we combined upd3null mutants with hemocyte-specific knockdown of mei-41 and tefu to test their susceptibility to oxidative stress. Our data indeed provide evidence that loss of upd3 rescues the higher susceptibility of flies with hemocyte-specific knockdown for tefu and mei-41 (Figure 6F). Furthermore, we see that upd3null mutants show a diminished susceptibility to oxidative stress compared to control flies (Figure 6F).

1. To link the observed effects to systemic metabolic changes, it would be useful to measure glycogen and triglycerides in these flies as well:crq-Gal80ts>reaper to see what role hemocytes play in the observed metabolic changes.Hml-Upd3 overexpression and Upd3 null mutant (Upd3 RNAi seems to be problematic, we have similar experiences) to see if Upd3 overexpression leads to even more profound changes as suggested, and if Upd3 mutation at least partially suppresses the observed changes.

We agree with the referee that analyzing the connection of hemocyte activation to metabolic changes should be demonstrated in our manuscript to support our claim that hemocytes are important regulators of energy mobilization during oxidative stress. Hence, we analyzed triglycerides and carbohydrate levels in hemocyte-deficient flies (crq-Gal80ts>reaper) during oxidative stress. Indeed, we found substantial differences in energy mobilization in these flies supporting the assumption that the higher susceptibility of hemocyte-deficient flies could be caused by substantial decrease in free glucose and inefficient lysis of triglycerides from the fat body (Figure 1I-K).

1. To test whether the cause of the increased susceptibility to oxidative stress is due to Upd3 overactivation induced by DDR silencing, the authors should attempt to rescue DDR silencing with an Upd3 null mutation.

The suggestion of the reviewer was included in the revised manuscript and as outlined above we now added this data set to our manuscript (Figure 6F). Indeed, we can now provide evidence that upd3null mutation rescues the higher susceptibility of flies with DDR knockdown in hemocytes.

1. Lethality after PQ treatment varies widely (sometimes from 10 to 90%! as in Figure 5D) - is this normal? In some experiments the variability was much lower. In particular, Figure 5D is very problematic and for example the result with upd3 null mutant compared to control is not very convincing. This could be an important result to test whether Upd3, with normal expression likely coming from cluster 6, actually plays a beneficial role, whereas overexpression with Hml leads to pathology.

We agree with the referee that it would be more convincing if the variation cross of survival experiments would be less. However, we included a lot of flies and vials in many individual experiments to test our hypothesis and variation in these survivals was always the case. These effects can be caused by many factors for example the amount of food intake by the flies, genetic background or inserted transgenes. The n-number is quite high across our survivals; so that we are convinced, the seen effects are valid. This reflects also the power of using *Drosophila melanogaster* as a model organism for such survivals. The high n-number in our data falls into a normal Gauss distribution with a distinct mean susceptibility between the genotypes analyzed.

1. I like the conclusion at the end of the results: line 413: "We show that this oxidative stressmediated immune activation seems to be controlled by non-canonical DNA damage signaling resulting in JNK activation and subsequent upd3 expression, which can render the adult fly more susceptible to oxidative stress when it is over-activated." This is actually a more appropriate conclusion, but in the summary, introduction and discussion along with the overall schematic illustration, this is not actually stated as such, but rather as Upd3 released from cluster 6 causes the pathology. For example: line 435 "Hence, we postulate that hemocyte-derived upd3, most likely released by the activated plasmatocyte cluster C6 during oxidative stress in vivo and subsequently controlling energy mobilization and subsequent tissue wasting upon oxidative stress."

We thank the referee for this suggestion and edited our manuscript and conclusions accordingly.

**Reviewer #3 (Recommendations For The Authors):**
1. In Figure 2, the authors claim showed that PQ treatment changes the hemocyte clusters in a way that suppresses the conventional Hml+ or Pxn+ hemocytes (cluster1) while expanding hemocyte clusters enriched with metabolic genes such as Lpin, bmm etc. It is not clear whether these cells are comparable to the fat body and if these clusters express any of previously known hemocyte marker genes to claim that these are bona fide hemocytes.

We now included a new analysis of our snRNA-seq data in Figure S4, where we clearly show that all identified hemocyte clusters do not have a fat body signature and are hemocytes, which seem to undergo metabolic adaptations (Figure S4A). Furthermore, we show that the identified fat body cells have a clear fat body signature (Figure S4B) and do not express specific hemocyte markers (Figure S4C).

1. In Figure 4C, the authors showed that comet assays of isolated hemocytes result in a statistically significant increase in DNA damage in DDR-deficient flies before and after PQ treatment. However, the authors conclude that, in lines 324-328, the higher susceptibility of DDR-deficient flies is not due to an increase in DNA damage. To explicitly conclude that "non-canonical" DNA damage response, without any DNA damage, is specifically upregulated during PQ treatment, the authors require further support to exclude the potential activation of canonical DDR.

The referee is correct that we do not provide direct evidence for non-canonical DNA damage signaling. Therefore, we also decided to tune down our statement here a bit and removed that claim from the title. Increase in DNA damage can of course also increase the non-canonical DNA damage signaling pathway, loss of DNA damage signaling genes such as tefu and mei-41 seem to only have minor impacts on the overall amount of DNA damage acquired in hemocytes by oxidative stress. We therefore concluded that the induction in immune activation is most unlikely only caused by increased DNA damage but might be connected to dysregulation in non-canonical DNA damage signaling. Canonical DNA damage signaling leads essentially to DDR, which could be slow in adult hemocytes because they post-mitotic, or to apoptosis, which we could not observe in the analyzed time window in our experiments. Hemocyte number remained stable over the 24h PQ treatment without reduction in cell number (Figure 1H).

1. From Figure 4D-F, the authors showed that loss of DDR in hemocytes induces the expression of unpaired 2 and 3, Socs36E, which represent the JAK/STAT pathway, and thor, InR, Pepck in the InR pathway, and a JNK readout, puc. These results indicate that the DDR pathway normally inhibits the upd-mediated JAK/STAT activation upon PQ treatment, compared to wild-type animals during PQ treatment in Figure 1B-C, which in turn protects the animal during oxidative stress responses. However, the authors claim that "enhanced DNA damage boosts immune activation and therefore susceptibility to oxidative stress (lines 365-366); we show that this oxidative stress-mediated immune activation seems to be controlled by non-canonical DNA damage signaling resulting in JNK activation and subsequent upd3 expression (line 413-416)". These conclusions are not compatible with the authors' data and may require additional data to support or can be modified.

In the revised manuscript, we carefully revised now the text and our statements that it seems that DNA damage signaling in hemocytes has regulatory or modulatory effect on the immune response during oxidative stress. Accordingly, we also adjusted our graphical summary. We agree with the referee and used the term “non-canonical” DNA damage signaling more carefully throughout the manuscript. The slight increase in DNA damage seen after PQ treatment can contribute to immune activation but seems to be not correlative to the induced cytokine levels or the susceptibility of the flies to oxidative stress.

1. In Fig 1I, the authors showed that genetic ablation of hemocytes using UAS-repear induces susceptibility to PQ treatment. It is possible that inducing cell death in hemocytes itself causes the expression of cytokine upd3 or activates the JNK pathway to enhance the basal level of upd3/JNK even without PQ treatment. If this phenotype is solely mediated by the loss of hemocytes, the results should be repeated by reducing the number of hemocytes with alternative genetic backgrounds.

In the different genotypes analyzed across our manuscript we did not detect cell death of hemocytes or a dramatic reduction in hemocytes number (see Figure 1H, Figure 5B, Figure 6C). The higher susceptibility if hemocyte-deficient flies during oxidative stress is most likely caused by the loss of their regulatory role during energy mobilization. We tested triglyceride levels in hemocyte-deficient flies and found a decreased triglyceride consumption (lipolysis), with reduced levels of circulating glucose levels. This findings support our hypothesis that hemocytes are needed to balance the response to oxidative stress. In contrast, the flies with DDR-deficient hemocytes show higher systemic cytokine levels, which most likely enhance energy mobilization from the fat body and therefore result in a higher susceptibility of the fly to oxidative stress. Hence, we claim that hemocytes and their regulation of systemic cytokine levels are important to balance the response to oxidative stress and guarantee the survival of the organism.

1. Lethality of control animals in PQ treatment is variable and it is hard to estimate the effect of animal susceptibility during 15mM PQ feeding. For example, Fig1A shows that control animals exhibit ~10% death during 15mM PQ which is further enhanced by crq-Gal80>reaper expression to 40% (Fig 1I). However, in Fig 5D-E, the basal lethality of wild-type controls already reaches 40~50%, which makes them hard to compare with other genetic manipulations. Related to this, the authors demonstrated that the expression of upd3 in hemocytes is sufficient to aggravate animal survival upon PQ treatment; however, upd3 null mutants do not rescue the lethality, which indicates that upd3 is not required for hampering animal mortality. These data need to be revisited and analyzed.

As outlined above, we find the variability of susceptibility to oxidative stress across all of our experiments. This could be due to different effects such as food intake but also transgene insertion and genetic background. Crq-gal80ts>reaper flies are healthy, but show a shortened life span on normal food (Kierdorf et al., 2020) due to enhanced loss of proteostasis in muscles. We show in the revised manuscript that these flies have a higher susceptibility to oxidative stress and that this effect could be mediated by defects in energy mobilization and redistribution as shown by less triglyceride lysis from the fat body and decreasing levels in free glucose. This would explain the high mortality rate of these flies at 7 days after eclosion. Paraquat treatment (15mM) is a severe inducer of oxidative stress, which results in death of most flies when they are maintained for longer time windows on PQ food. Hence, it is a model, which is not suitable to examine and monitor recovery from this detrimental insult. upd3null mutants were extensively reexamined in this manuscript, and even though we could not see a full protection of these flies from oxidative stress induced death, we found a reduced susceptibility compared to control flies (Figure 6F). Furthermore, when we combined upd3null mutants with flies deficient for tefu and mei-41 in hemocytes, the increased susceptibility to oxidative stress was rescued.